# Storyboard-guided Alignment for Fine-grained Video Action Recognition

**Enqi Liu**[1], **Liyuan Pan**[1,2]*, **Yan Yang**[3], **Yiran Zhong**[4], **Zhijing Wu**[1], **Xinxiao Wu**[1], **Liu Liu**[5]

[1]Beijing Institute of Technology, Beijing, China
[2]Yangtze Delta Region Academy of Beijing Institute of Technology, Jiaxing, China
[3]BDSI, Australian National University, Canberra, Australia
[4]OpenNLPLab, Shanghai, China
[5]Huawei, Beijing, China
{enqi.liu, liyuan.pan, zhijingwu, wuxinxiao}@bit.edu,
yan.yang@anu.edu.au, zhongyiran@gmail.com, liuliu33@huawei.com

## Abstract

Fine-grained video action recognition can be formulated as a video–text matching problem. Previous approaches primarily rely on global video semantics to consolidate video embeddings, often leading to misaligned video–text pairs due to inaccurate atomic-level action understanding. This inaccuracy arises due to i) videos with distinct global semantics may share similar atomic actions or visual appearances, and ii) atomic actions can be momentary, gradual, or not directly aligned with overarching video semantics. Inspired by storyboarding, where a script is segmented into individual shots, we propose a multi-granularity framework, SFAR. SFAR generates fine-grained descriptions of common atomic actions for each global semantic using a large language model. Unlike existing works that refine global semantics with auxiliary video frames, SFAR introduces a filtering metric to ensure correspondence between the descriptions and the global semantics, eliminating the need for direct video involvement and thereby enabling more nuanced recognition of subtle actions. By leveraging both global semantics and fine-grained descriptions, our SFAR effectively identifies prominent frames within videos, thereby improving the accuracy of embedding aggregation. Extensive experiments on various video action recognition datasets demonstrate the competitive performance of our SFAR in supervised, few-shot, and zero-shot settings.

## 1   Introduction

Fine-grained video action recognition has garnered increasing attention due to its broad applicability in areas such as sports analytics [54], human-computer interaction [13], surveillance [44], and video understanding [12]. In contrast to standard action recognition, fine-grained video action recognition necessitates a more detailed understanding of actions with similar appearances, demanding greater precision in capturing prominent frames with subtle actions.

The advent of large language models has revealed that a robust multimodal encoder like CLIP [30] can consolidate significantly more potent learned embeddings compared to manually crafted embeddings for action recognition [39, 21, 29, 2, 23, 28, 51, 15, 11, 47]. These methods involve using CLIP's visual encoder to extract video embeddings and its textual encoder to extract text embeddings. The video and text embeddings are then aligned for video classification. However, the potential of the textual encoder has not been fully explored.

---

*Corresponding author.

39th Conference on Neural Information Processing Systems (NeurIPS 2025).

To effectively integrate textual information, several works [48, 26, 8, 16, 24] refine global semantics by leveraging LLMs/MLLMs or additional lexicons to generate class-specific atomic action descriptions. These auxiliary descriptions are selected based on their alignment with each video frame, which leads to two rigid assumptions: i) each global video semantic must correspond to identical atomic actions, and ii) all atomic actions within a video should be closely related to its global semantics. However, these assumptions are frequently violated, leading to inaccurate embeddings. For instance, in Fig. 1 (a), the videos 'Baking cookies' and 'Making pizza' are ambiguous with each other, featuring overlapping atomic actions. In Fig. 1 (b), the atomic action 'Stand' can be irrelevant to the global video semantics 'Swing legs'. Moreover, the non-uniform distribution nature of

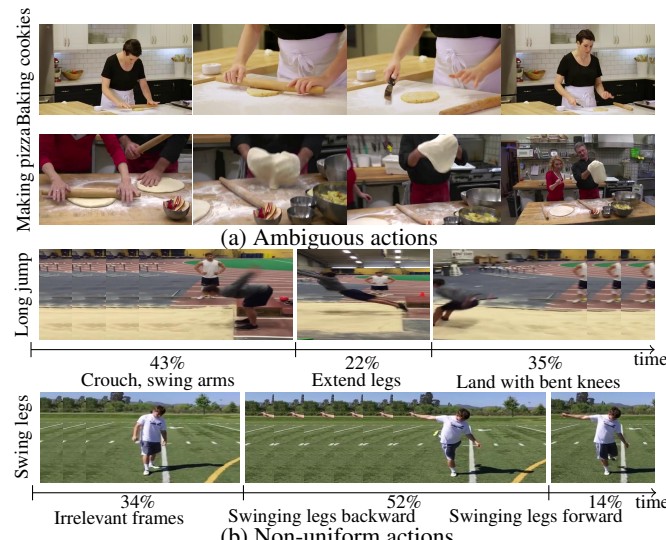

(a) Ambiguous actions

(b) Non-uniform actions

Figure 1: *Example videos with ambiguous or non-uniform actions. Each row displays sample frames of a video with its class name. (a) Ambiguous actions. Both 'Baking cookies' and 'Making pizza' are actions performed in the kitchen with similar visual appearances and share atomic actions,* e.g.*, the 'rolling dough'. (b) Non-uniform actions. Multiple atomic actions distribute unevenly within a video to form an action,* e.g.*, the video 'Long jump' is composed of 43% of 'Crouch, swing arms' and 22% of 'Extend legs'. Moreover, some atomic actions may be irrelevant to the global class,* e.g.*, 34% of the frames show 'Stand', unrelated to 'Swing Legs'. The axis indicates the percentage of frames per atomic action.*

atomic actions has been overlooked previously, *i.e.*, the duration of each atomic action can vary. Based on this, using the global video semantics directly for aligning videos and text prompts can lead to misunderstandings due to their granularity discrepancy. In this paper, we address these issues and propose a multi-granularity framework, SFAR. The core of our SFAR is decomposing 'the global video semantics into fine-grained sub-texts' and 'a video action into multiple atomic actions'. This allows both coarse and fine-grained identifications of prominent frames in videos.

Inspired by the concept of storyboarding, which breaks down a script into individual shots, we enhance the global semantics (text) by generating detailed descriptions using a pre-trained large language model (LLM). These fine-grained descriptions, *i.e.*, sub-texts, capture common atomic actions depicted in videos, requiring only their class names and utilizing our designed question prompts, eliminating the need for direct video involvement. To fit the flexibility of various pre-trained LLM and question prompt formats, we design a text prompt perplexity metric to measure the diversity among sub-texts and the similarity between the sub-text and the global text for filtering sub-texts. It provides an effective schema for selecting sub-texts to train the video action recognition model.

We then use global texts and sub-texts to coarsely and fine-grainedly compute a video embedding for videos with ambiguous or non-uniform actions. Specifically, we augment the global text with sub-texts in the embedding space of CLIP by abounding described video actions with sub-texts, which also decreases the granularity of the global text. This augmented global text is used to weight video frames in embedding space and compute a coarse video embedding. We illustrate network attention of the augmented global text in Fig. 2 (c) and sub-texts in Fig. 2 (d). Compared to the previous method constrained by granularity differences (Fig. 2 (b)), our augmented global text and sub-texts direct network attention more comprehensively toward regions with actions.

We then fuse the coarse and fine-grained video embedding to compute a video embedding that improves the classification performance on ambiguous and non-uniform videos. Heatmaps of our network are in Fig. 2 (e), and improvements over videos with different ambiguous and non-uniform scores are in Fig. 2 (f). We rigorously validate our approach across different scenarios, including supervised, few-shot, and zero-shot video action recognition. Our method delivers top-notch performance in all these scenarios, showcasing the effectiveness of our framework.

Our main contributions are:

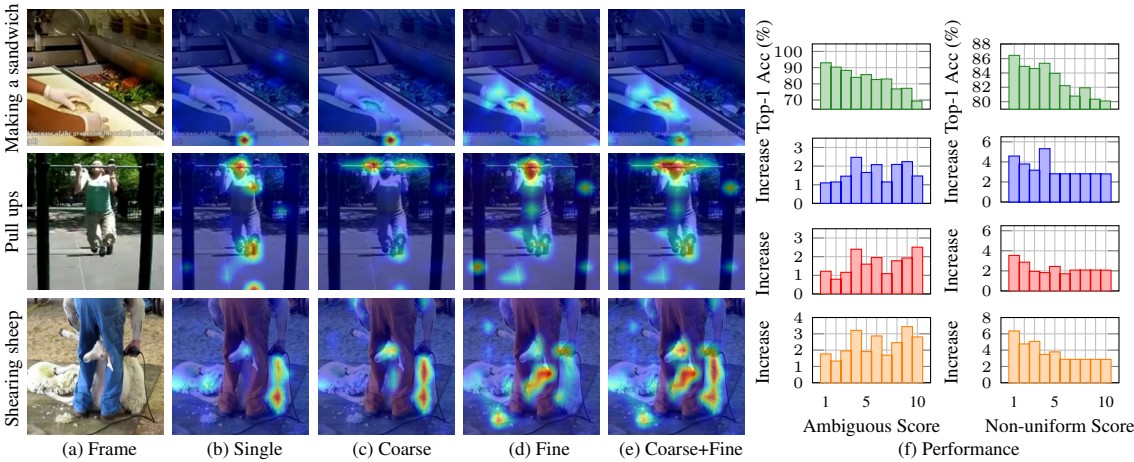

Figure 2: *Examples that illustrate the ineffectiveness of directly aligning video frames and global text, from (a-e) heatmaps and (f) classification performance improvements. (a) Video frames with their class name annotated at the left. (b) Heatmaps of the single granularity-based (single) method [48], which are limited by the granularity difference between the video frame and global text. (c-e) Heatmaps using coarse video embedding, fine-grained video embedding, and the fusion of the two. (f) We show the performance distributions (Top-1 Accuracy) over ambiguous scores and non-uniform scores of videos for (b-e) on the whole set. The first row is the absolute performance for (b), and the last three rows are improvements of (c), (d), and (e) concerning (b). Please refer to our supplementary materials for details on the computation of the ambiguous score and the non-uniform score, as well as the accuracy improvements on the Kinetics-400 dataset presented in Tab. 8.*

- A multi-granularity framework for transferring CLIP trained on image-text pair to video action recognition.
- A schema for automatically decomposing a video action into common atomic actions to provide fine-grained knowledge of the video action to CLIP.
- A coarse and fine-grained video embedding module for videos with ambiguous and non-uniform actions.

## 2 Related Works

We review video action recognition methods focusing visual and visual-text representation learning.

**Visual Representation Learning.** Action recognition requires accurately capturing temporal semantic variations. Numerous studies [59, 52, 42, 35, 39, 55] have thus focused on visual representation learning. Early works studied the joint learning of spatial and temporal features of a video using various architectures [26, 23, 2, 32]. Benefiting from large-scale visual and visual-text pre-training, recent methods leverage the strong spatial features learned in pre-training and focus on fine-tuning the trained model to capture the temporal semantics of a video. However, these methods do not explicitly address the fundamental challenge of recognizing video actions from ambiguous and non-uniform videos. This paper proposes decomposing video actions into atomic actions to improve the recognition of challenging videos.

**Visual-text Representation Learning.** Several methods [8, 40, 43, 47, 48, 41] have been developed to overcome these limitations by identifying video frames that strongly align with the text prompts of video actions for video action recognition in vision-language models, *e.g.*, CLIP. One remarkable work is BIKE [48], which weights the video frames based on their alignment with the text prompts to compute a video embedding for action recognition. However, these methods are limited by the granularity difference between the video frames and the text prompts, where the text prompt is a global context for the video. Unlike previous methods focused on generative modeling [49, 25, 57], where LLMs generate captions or texts conditioned on video or frame embeddings, our approach adopts a discriminative modeling approach. We generate sub-texts from the global text to describe atomic actions in video frames. The sub-texts are then used to construct coarse and fine-grained video embeddings, which improves the recognition of fine-grained actions.

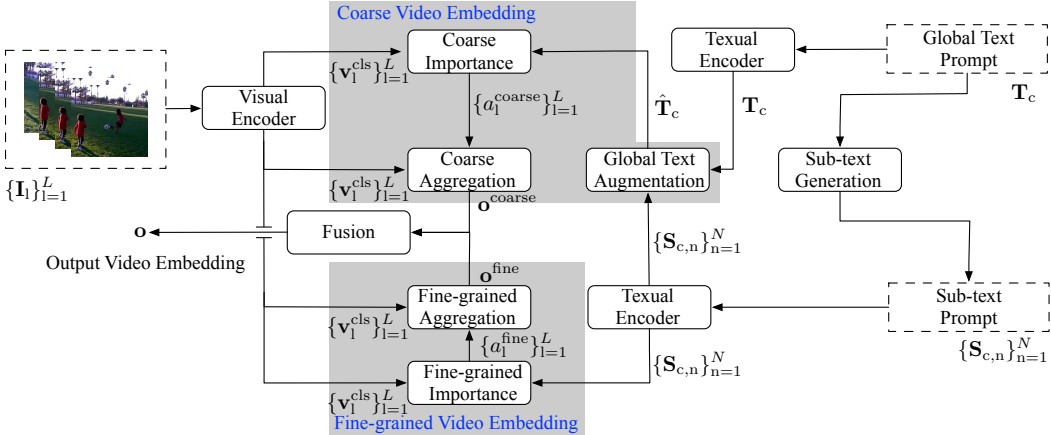

Figure 3: *An overview of our framework for video action recognition. We extend CLIP for classifying the video $\{\mathbf{I}_l\}_{l=1}^L$ with $L$ frames by computing a video embedding from frame embedding $\{\mathbf{v}_l^{\text{cls}}\}_{l=1}^L$ in three steps. (i) We decompose the global text prompt $\mathbf{T}_c$ that describes the class semantic of action c into descriptions of atomic action (i.e., sub-text prompts) by using a pre-trained large language model. The global text prompt $\mathbf{T}_c$ and sub-text prompts are then embedded by the textual encoder of CLIP for extracting embeddings of $\mathbf{T}_c$ and $\{\mathbf{S}_{c,n}\}_{l=1}^L$. (ii) A coarse video embedding is extracted by augmenting the global text embedding $\mathbf{T}_c$ with the sub-text embedding $\{\mathbf{S}_{c,n}\}_{l=1}^L$, calculating coarse importance of the frame embedding $\{\mathbf{v}_l^{\text{cls}}\}_{l=1}^L$ with the augmented global text embedding $\hat{\mathbf{T}}_c$, and using the importance $\{a_l\}_{l=1}^L$ to aggregate a coarse video embedding $\mathbf{o}^{\text{coarse}}$ from frame embedding $\{\mathbf{v}_l^{\text{cls}}\}_{l=1}^L$. (iii) Similar to the coarse video embedding, we get a fine-grained video embedding $\mathbf{o}^{\text{fine}}$ by calculating fine-grained importance $\{a_l^{\text{fine}}\}_{l=1}^L$ of the frame embedding $\{\mathbf{v}_l^{\text{cls}}\}_{l=1}^L$ with sub-text embedding $\{\mathbf{S}_{c,n}\}_{l=1}^L$ for aggregating frame embedding $\{\mathbf{v}_l^{\text{cls}}\}_{l=1}^L$. The coarse and fine-grained embeddings $\mathbf{o}^{\text{coarse}}$ and $\mathbf{o}^{\text{fine}}$ are fused to form the final video embedding $\mathbf{o}$ for action recognition.*

## 3 Methodology

**Preliminary.** CLIP [30], a visual-language pre-training method, consists of a visual and a textual encoder. It learns a joint embedding space by maximizing similarities between aligned image-text pairs and minimizing them for misaligned ones. Given an image $\mathbf{I}$ and global text prompts $\{\mathbf{T}_c\}_{c=1}^C$ of $C$ classes formatted as [a photo of a class] that globally describe class semantics, where class is the class name, CLIP performs zero-shot classification by extracting the visual class embedding $\mathbf{v}^{\text{cls}}$ from $\mathbf{I}$ and text class embeddings $\{\mathbf{t}_c^{\text{cls}}\}_{c=1}^C$ from $\{\mathbf{T}_c\}_{c=1}^C$. The image $\mathbf{I}$ is then classified into the class c' with the maximum cosine similarity, *i.e.*, $c' = \arg\max_c \text{sim}(\mathbf{v}^{\text{cls}}, \mathbf{t}_c^{\text{cls}})$.

**Overview.** To extend CLIP for classifying the video $\{\mathbf{I}_l\}_{l=1}^L$ with $L$ frames, we adaptively compute a coarse and a fine-grained video embedding. Two multi-granularity embeddings capture global semantic and atomic semantics of ambiguous and non-uniform actions in the video, for computing the cosine similarity with the text class embedding. There are three key steps:

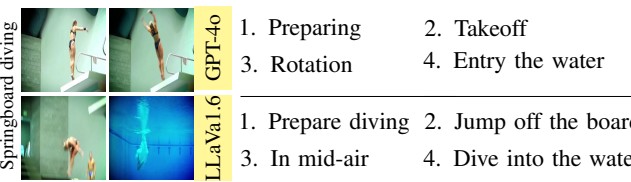

Figure 4: *Example sub-texts $\{\mathbf{S}_{c,n}\}_{n=1}^N$ generated from GPT-4o and LLaVa1.6, with $N = 4$ in the example.*

(i) Sub-text generation, decomposing each global text prompt $\mathbf{T}_c$ into a sequence of $N$ atomic action text descriptions $\{\mathbf{S}_{c,n}\}_{n=1}^N$ (*i.e.*, sub-text prompts), by using a pre-trained large language model; (ii) Coarse video embedding, embedding each frame $\mathbf{I}_l$ into a visual class embedding $\mathbf{v}_l^{\text{cls}}$ that is a frame embedding. Then, by using the global text prompt augmented with sub-text prompts to identify salient video frames in CLIP embedding space, we aggregate the frame embedding $\{\mathbf{v}_l^{\text{cls}}\}_{l=1}^L$ into coarse video embeddings $\mathbf{o}^{\text{coarse}}$; (iii) Fine-grained video embedding, finding video frames with atomic actions by using sub-text prompts and CLIP for computing a fine-grained video embedding $\mathbf{o}^{\text{fine}}$. We fuse the coarse and fine-grained video embedding into a video embedding $\mathbf{o}$ to compute

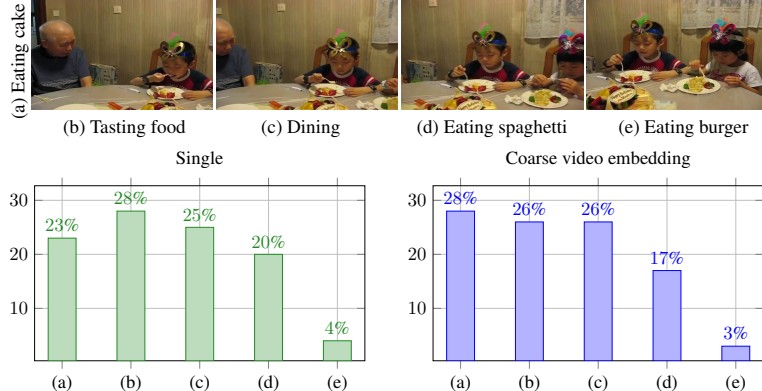

Figure 5: *An example illustrating the benefits of using our coarse video embedding for ambiguous actions. We show a video in the first row, with the ground truth action labeled as (a) on the left. Its ambiguous actions are annotated with (b), (c), (d) and (e) at the bottom of the video. The action probability from the single granularity-based method [48] and our coarse video embedding are in the next row. Compared with the single granularity-based method that identifies the video as (b) 'Tasting food', we obtain a higher probability of the correct class (a) 'Eating cake'.*

cosine similarities with text class embeddings $\mathbf{t}_c^{cls}$ for classifying the video. The overview of our method is in Fig. 3.

### 3.1 Sub-text Generation

A global text prompt $\mathbf{T}_c$ is a description of a video action. It overlooks that an action is usually formed by performing a sequence of atomic actions, like a video performing the action. Directly aligning video frame embedding to the global text prompt $\mathbf{T}_c$ introduces the granularity differences and decreases the video classification performance. To minimize human burden and bias, we propose a pipeline that automatically generates and selects sub-texts based on the global text (class name).

We leverage a pre-trained large language model $\text{LLM}(\cdot)$ to decompose global text prompt $\mathbf{T}_c$ into $N$ potentially atomic actions $\{\mathbf{S}_{c,n}\}_{n=1}^N = \text{LLM}(\mathbf{P}_c)$, where $\mathbf{P}_c$ is the prompt for describing the action class c and directing the pre-trained large language model to find potential atomic action descriptions, sub-texts $\{\mathbf{S}_{c,n}\}_{n=1}^N$. Refer to Fig. 4 for an example.

However, the existence of various pre-trained large language models and the flexibility in designing $\mathbf{P}_c$ make selecting a meaningful sub-text set challenging. We assume that an optimal sub-text set $\{\mathbf{S}_{c,n}\}_{n=1}^N$ for video classification should be sufficiently related with the global text prompt $\mathbf{T}_c$ while each sub-text should be diverse from the others. To measure the similarities and diversity, we propose a text prompt perplexity metric to select the sub-text set with the highest text prompt perplexity score.

In Fig. 5, we show ambiguous action probability computed from a single granularity-based method [48] and our coarse video embedding. Our method uses sub-text to augment the global text help the model to identify different atomic actions in ambiguous actions, and improve the accuracy of recognizing ambiguous actions.

With the class embedding of the global text $\mathbf{T}_c$ and sub-texts $\{\mathbf{S}_{c,n}\}_{n=1}^N$ from the CLIP textual encoder, $\mathbf{t}_c^{cls}$ and $\mathbf{s}_{c,n}^{cls}$, we define the text prompt perplexity score $\text{TPP}_c$ for $\mathbf{t}_c^{cls}$ and $\mathbf{s}_{c,n}^{cls}$ as below

$$\text{TPP}_c = \exp\left(-\frac{1}{N}\sum_{n=1}^N \log\left(\alpha(\sigma_{c,n})\beta(\delta_{c,n})\right)\right), \tag{1}$$

$$\sigma_{c,n} = \frac{\text{sim}(\mathbf{t}_c^{cls}, \mathbf{s}_{c,n}^{cls}) + 1}{2}, \tag{2}$$

$$\delta_{c,n} = 1 - \frac{1}{N-1}\sum_{n'=1,n'\neq n}^N \frac{\text{sim}(\mathbf{s}_{c,n}^{cls}, \mathbf{s}_{c,n'}^{cls}) + 1}{2}, \tag{3}$$

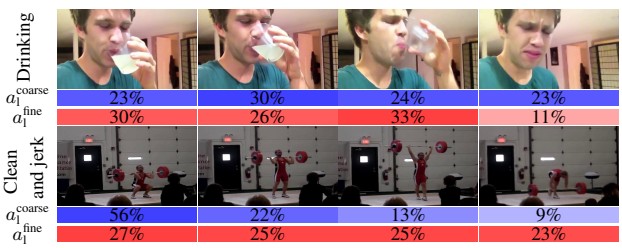

Figure 6: *Examples of coarse and fine-grained importance probability for videos with non-uniform actions. The fine-grained importance probability $a_l^{\text{fine}}$ allocate larger scores than the coarse importance probability $a_l^{\text{coarse}}$ to prominent frames. For example, the illustrated fourth frame in the second row, 'lowering barbell back down to the ground', represents an atomic action of 'Clean and jerk', and $a_l^{\text{fine}} = 23\%$ has a higher probability than $a_l^{\text{coarse}} = 9\%$.*

where the $\text{sim}(\cdot, \cdot)$ calculates the cosine similarities of inputs. We employ $\sigma_{c,n}$ to quantify the similarity between a global text and its corresponding sub-texts, a higher value of $\sigma_{c,n}$ indicates greater similarity. Meanwhile, $\delta_{c,n}$ is used to evaluate the similarity among sub-texts themselves, where a higher value of $\delta_{c,n}$ indicates greater diversity across the sub-texts. Here, $\alpha(\cdot)$ and $\beta(\cdot)$ are linear scaling functions for the scores. Please refer to Fig. 7b for the correlations between the video classification performance and the text prompt perplexity score $\text{TPP}_c$.

## 3.2 Coarse Video Embedding

For identifying different atomic actions in ambiguous videos, we augment the global text $\mathbf{T}_c$ with the sub-texts $\{\mathbf{S}_{c,n}\}_{n=1}^N$ to adaptively aggregate the video frames $\{\mathbf{I}_l\}_{l=1}^L$ in CLIP embedding space. As the class name in global text is usually a phrase, and sub-text $\mathbf{S}_{c,n}$ describes the atomic action with multiple words, we keep the word embeddings to benefit from the rich semantics of each word embedding for video frame aggregations. We denote the matrix form of the word embeddings for global text and sub-text as $\mathbf{T}_c$ and $\mathbf{S}_{c,n}$, and augment $\mathbf{T}_c$ by the cross-attention mechanism,

$$\mathbf{Q}_c = \mathbf{T}_c \mathbf{W}^q, \quad \mathbf{K}_{c,n} = \mathbf{S}_{c,n} \mathbf{W}^k, \quad \mathbf{V}_{c,n} = \mathbf{S}_{c,n} \mathbf{W}^v, \tag{4}$$

$$\mathbf{K}_c = [\{\mathbf{K}_{c,n}\}_{n=1}^N], \quad \mathbf{V}_c = [\{\mathbf{V}_{c,n}\}_{n=1}^N], \tag{5}$$

$$\hat{\mathbf{T}}_c = \text{Attention}(\mathbf{Q}_c, \mathbf{K}_c, \mathbf{V}_c) + \mathbf{T}_c, \tag{6}$$

where $\mathbf{W}^q$, $\mathbf{W}^k$, and $\mathbf{W}^v$ are matrices for projecting the query $\mathbf{Q}_c$, key $\mathbf{K}_{c,n}$, and value $\mathbf{V}_{c,n}$. Here, $[\{\mathbf{K}_{c,n}\}_{n=1}^N]$ and $[\{\mathbf{V}_{c,n}\}_{n=1}^N]$ concatenates all keys and values for $\mathbf{Q}_c$.

Using the augmented global text embeddings $\hat{\mathbf{T}}_c$, we find the salient video frame embedding $\mathbf{v}_l^{\text{cls}}$ for aggregating a coarse video embedding. To find the coarse-grained importance score $a_l^{\text{coarse}}$ of each frame embedding, we calculate the overall normalized similarity between each word embedding and the video frame embedding by

$$a_l^{\text{coarse}} = \sum_{\hat{\mathbf{t}}_c \in \hat{\mathbf{T}}_c} \frac{\exp\left(\text{sim}(\hat{\mathbf{t}}_c, \mathbf{v}_l^{\text{cls}})\right)}{\sum_{l'=1}^L \exp\left(\text{sim}(\hat{\mathbf{t}}_c, \mathbf{v}_{l'}^{\text{cls}})\right)}. \tag{7}$$

We then compute the coarse video embedding $\mathbf{o}^{\text{coarse}}$ with summing over the video frame embedding $\mathbf{v}_l^{\text{cls}}$ weighted by $a_l^{\text{coarse}}$,

$$\mathbf{o}^{\text{coarse}} = \sum_{l=1}^L \mathbf{v}_l^{\text{cls}} a_l^{\text{coarse}}. \tag{8}$$

## 3.3 Fine-grained Video Embedding

While the video coarse embedding $\mathbf{o}^{\text{coarse}}$ captures salient semantics for ambiguous actions, it may overlook atomic actions in the video that align with the expected action class c due to the non-uniform property of video actions. To address the limitation, we use the word embedding of the sub-texts to find atomic actions among the video frame embeddings $\{\mathbf{v}_l^{\text{cls}}\}_{l=1}^L$ to compute the fine-grained video embedding.

Table 1: *Comparison result on the Kinetics-400 dataset. We report the GFLOPs in the inference phase. "Views" indicates the number of temporal clips and spatial crops used in inference (# temporal clip × # spatial crop). The magnitude is Million ($10^6$) for number of model parameters (Param). We achieve the highest top-1 and top-5 accuracy by employing only an 8-frame input evaluated with a single view, outperforming past methods with more model parameters, input frames, and views. We highlight the best top-1 and top-5 accuracy in* **bold**.

| Method | Venue | Input | Pre-training | Top-1(%) | Top-5(%) | Views | GFLOPs | Param |
|---|---|---|---|---|---|---|---|---|
| *Methods with large-scale visual pre-training* | | | | | | | | |
| MVFNet$_{En}$ [45] | AAAI'21 | 24×224$^2$ | ImageNet-1K | 79.1 | 93.8 | 10×3 | 188×30 | - |
| ViViT-B/16×2 [2] | ICCV'21 | 32×224$^2$ | JFT-300M | 80.0 | - | 4×1 | 455.2 | 151.9 |
| ST-Adapter ViT-B/16 [28] | NeurIPS'22 | 8×224$^2$ | WIT-400M | 82.0 | 95.7 | 3×1 | 455 | 128.8 |
| EVL ViT-B/16 [21] | ECCV'22 | 8×224$^2$ | WIT-400M | 82.9 | - | 3×1 | 444 | 177.7 |
| MTV-B [50] | CVPR'22 | 32×224$^2$ | JFT-300M | 81.8 | 95.0 | 4×3 | 384×12 | 310 |
| VideoSwin-B [23] | CVPR'22 | 32×224$^2$ | ImageNet-21K | 82.7 | 95.5 | 4×3 | 282×12 | 88.1 |
| ActionCLIP ViT-B/16 [39] | TNNLS'23 | 16×224$^2$ | WIT-400M | 82.6 | 96.2 | 10×3 | 282×30 | 141.7 |
| ATM ViT-B/16 [46] | ICCV'23 | 8×224$^2$ | WIT-400M | 82.8 | 95.6 | 1×1 | 378×1 | - |
| DIST ViT-B/16 [29] | ICCV'23 | 8×224$^2$ | WIT-400M | 83.6 | - | 3×1 | 163×3 | 105 |
| AIM ViT-B/16 [53] | ICLR'23 | 8×224$^2$ | WIT-400M | 83.9 | 96.3 | 3×1 | 202×3 | 97 |
| ILA-ViT-B/16 [37] | ICCV'23 | 8×224$^2$ | WIT-400M | 84.0 | 96.6 | 4×3 | 149×12 | - |
| *Methods with large-scale visual-text pre-training* | | | | | | | | |
| X-CLIP ViT-B/16 [26] | ECCV'22 | 8×224$^2$ | WIT-400M | 82.3 | - | 1×1 | 145×1 | - |
| VideoPrompt ViT-B/16 [15] | ECCV'22 | 16×224$^2$ | WIT-400M | 76.9 | 93.5 | 1×5 | - | 154 |
| SIF ViT-B/16 [40] | ACMMM'23 | 8×224$^2$ | WIT-400M | 77.4 | 93.6 | 4×3 | 1136×12 | 143.9 |
| Vita-CLIP ViT-B/16 [43] | CVPR'23 | 8×224$^2$ | WIT-400M | 80.5 | 96.0 | 1×1 | 97×1 | 187.9 |
| Vita-CLIP ViT-B/16 [43] | CVPR'23 | 8×224$^2$ | WIT-400M | 81.8 | 96.0 | 4×3 | 97×12 | 187.9 |
| BIKE ViT-B/16 [48] | CVPR'23 | 8×224$^2$ | WIT-400M | 83.2 | - | 1×1 | - | 124.1 |
| BIKE ViT-B/16 [48] | CVPR'23 | 8×224$^2$ | WIT-400M | 83.9 | - | 4×3 | - | 124.1 |
| M$^2$-CLIP ViT-B/16 [38] | AAAI'24 | 8×224$^2$ | WIT-400M | 83.4 | 96.3 | 4×3 | 214×3 | 165 |
| ALT ViT-B/16 [7] | CVPR'24 | 8×224$^2$ | WIT-400M | 83.6 | 95.9 | 3×1 | 328 | 134.4 |
| OST ViT-B/16 [6] | CVPR'24 | 8×224$^2$ | WIT-400M | 82.0 | - | 1×1 | - | - |
| **Ours ViT-B/16** | NeurIPS'25 | 8×224$^2$ | WIT-400M | **84.5** | **96.7** | 1×1 | 90.2×1 | 126.1 |

We compute a fine-grained importance score $a_l^{\text{fine}}$ for each video frame embedding $\mathbf{v}_l^{\text{cls}}$ as the maximum average similarity between $\mathbf{v}_l^{\text{cls}}$ and $\mathbf{S}_{c,n}$, and normalize it by a SoftMax function,

$$a_l^{\text{fine}} = \frac{\exp\left(\max_n \frac{1}{|\mathbf{S}_{c,n}|} \sum \mathbf{s}_{c,n} \in \mathbf{S}_{c,n} \text{sim}(\mathbf{s}_{c,n}, \mathbf{v}_l^{\text{cls}})\right)}{\sum_{l'=1}^{L} \exp\left(\max_n \frac{1}{|\mathbf{S}_{c,n}|} \sum \mathbf{s}_{c,n} \in \mathbf{S}_{c,n} \text{sim}(\mathbf{s}_{c,n}, \mathbf{v}_{l'}^{\text{cls}})\right)} , \tag{9}$$

where $|\mathbf{S}_{c,n}|$ is the number of words in the sub-text, and $\mathbf{s}_{c,n} \in \mathbf{S}_{c,n}$ is a word embedding for the sub-text. Similar to Eq. (8), we weight the video frame embedding $\mathbf{v}_l^{\text{cls}}$ with $a_l^{\text{fine}}$, and get the fine-grained video embedding $\mathbf{o}^{\text{fine}}$ by

$$\mathbf{o}^{\text{fine}} = \sum_{l=1}^{L} \mathbf{v}_l^{\text{cls}} a_l^{\text{fine}} . \tag{10}$$

We compare the coarse scores $a_l^{\text{coarse}}$ and fine-grained scores $a_l^{\text{fine}}$ in Fig. 6, and find that $a_l^{\text{fine}}$ can fine-grainedly allocate larger scores than $a_l^{\text{coarse}}$ to prominent frames that humans perform the action peak in the non-uniform videos.

### 3.4 Loss

We fuse the coarse video embedding and fine-grained video embedding as a video embedding $\mathbf{o}$ with two feedforward layers $\text{FFN}^{\text{coarse}}(\cdot)$ and $\text{FFN}^{\text{fine}}(\cdot)$, projecting the coarse video embedding and fine-grained video embedding to the semantic of class c,

$$\mathbf{o} = \text{FFN}^{\text{coarse}}(\mathbf{o}^{\text{coarse}}) + \text{FFN}^{\text{fine}}(\mathbf{o}^{\text{fine}}) , \tag{11}$$

and compute the cosine similarity with the text class embedding $\mathbf{t}_c^{\text{cls}}$ by $\text{sim}(\mathbf{t}_c, \mathbf{o})$.

We optimize our network by maximizing the similarity $y_{b,c^{\text{gt}}}$ between the b-th video embedding in a batch and text embedding of its ground truth class $c^{\text{gt}}$, and minimizing the similarity between other video and text class embeddings $\{y_{b,c}\}_{c=1,c\neq c^{\text{gt}}}^{C}$. Following [27], we use the InfoNCE loss,

$$\mathcal{L}_{\text{T2V}} = \frac{1}{B} \sum_{b=1}^{B} \frac{1}{|\mathbf{k}_b|} \sum_{b' \in \mathbf{k}_b} \log \frac{\exp(y_{b',c^{\text{gt}}})}{\sum_{b''=1}^{B} \exp(y_{b'',c^{\text{gt}}})} , \tag{12}$$

Table 2: *Comparative experiments are conducted on Charades [33], HMDB-51 [19], and UCF-101 [34], while ablation studies are performed on the Kinetics-400 dataset [17]. We report accuracy (%) for a single 8-frame clip with a spatial resolution of 224×224, unless otherwise specified. The "VZ" column denotes if the method is specifically designed for zero-shot action recognition or adapted from CLIP.*

| Method | Frames | mAP |
|---|---|---|
| MultiScale TRN [58] | - | 25.2 |
| STM [14] | 16 | 35.3 |
| SlowFast+NL [10] | 16+64 | 42.5 |
| X3D-XL(312) [9] | 16 | 43.4 |
| ActionCLIP [39] | 32 | 44.3 |
| BIKE [48] | 16 | 50.4 |
| **Ours** | 16 | **51.1** |

(a) *Comparisons for multi-label action recognition on the Charades dataset [33].*

| Method | VZ | UCF-101 | HMDB-51 |
|---|---|---|---|
| E2E [3] | ✓ | 44.1 | 29.8 |
| ER [5] | ✓ | 51.8 | 35.3 |
| ResT [20] | ✓ | 58.7 | 41.1 |
| Vita-CLIP [43] | ✗ | 75.0 | 48.6 |
| BIKE [48] | ✗ | 78.4 | 55.6 |
| M2-CLIP [38] | ✗ | 78.7 | 47.1 |
| **Ours** | ✗ | **79.0** | **56.6** |

(b) *Comparisons of zero-shot action recognition on the HMDB-51 [19] and UCF-101 [34] datasets.*

| Backbone | FVE | Top-1(%) | Top-5(%) |
|---|---|---|---|
| VideoPrompt [15] | ✗ | 76.9 | 93.5 |
| | ✓ | 79.3 | 95.1 |
| ATM [46] | ✗ | 82.8 | 95.6 |
| | ✓ | 82.9 | 96.4 |
| BIKE [48] | ✗ | 83.2 | 96.0 |
| | ✓ | 83.8 | 96.5 |

(c) *Generalization of fine-grained video embedding (FVE) on state-of-the-art methods using the Kinetics-400 dataset [17].*

| Generator | TPP | Top-1(%) | Top-5(%) |
|---|---|---|---|
| LLaVA-1.6-34b | 54.4 | 84.35 | 96.5 |
| GPT-Davinci | 54.7 | 84.35 | 96.6 |
| GPT-3.5 | 60.4 | 84.47 | 96.7 |
| GPT-4o | **60.8** | **84.52** | **96.7** |

(d) *Comparison of sub-texts generated from different LLMs on Kinetics-400. A higher Text Prompt Perplexity (TPP) score indicates better sub-texts.*

| Coarse VE | Fine-grained VE | Top-1(%) | Top-5(%) |
|---|---|---|---|
| Baseline (CLIP) | | 79.9 | 94.7 |
| Baseline (Temporal) | | 80.3 | 95.0 |
| ✗ | ✓ | 82.7 | 96.2 |
| ✓ | ✗ | 83.5 | 96.3 |
| ✓ | ✓ | **84.5** | **96.7** |

(e) *Ablations of coarse and fine-grained video embedding (VE) on Kinetics-400. Using fine-tuned CLIP and CLIP model with temporal layers as baselines.*

$$\mathcal{L}_{\text{V2T}} = \frac{1}{B} \sum_{b=1}^{B} \frac{1}{|\mathbf{k}_b|} \sum_{b' \in \mathbf{k}_b} \log \frac{\exp(y_{b', c^{gt}})}{\sum_{c=1}^{C} \exp(y_{b', c})} \ , \tag{13}$$

$$\mathcal{L} = \mathcal{L}_{\text{T2V}} + \lambda \mathcal{L}_{\text{V2T}} \ , \tag{14}$$

where $B$ is the number of batches, $\mathbf{k}_b$ find the index of the video that has the same class with the b-th video, $|\mathbf{k}_b|$ is its size, and $\lambda$ is a hyperparameter.

# 4 Experiment

## 4.1 Experimental Setup

Our proposed sub-text set and code are given in our project page.

**Datasets.** We experiment across four extensively recognized video benchmarks: Kinetics-400 [17], Charades [33], UCF-101 [34], and HMDB-51 [19] datasets.

**Supervised Learning.** Our model is implemented using the PyTorch framework. We train our network with batch size 256 for 30 epochs using the AdamW optimizer. The learning rate is set to $5 \times 10^{-5}$, and we use the cosine annealing strategy with 5 warm-up epochs. We follow [48] for data augmentation in training.

**Zero-shot Learning.** We evaluate our model, pre-trained on Kinetics-400 [17], using the UCF-101 [34] and HMDB-51 [19].

**Few-shot Learning.** We follow [39, 26] to experiment with different shot settings, selecting 2, 4, 8, and 16 examples per human action category for training. Training for 2 epochs on the Kinetics-400 dataset, and then use the same settings to train on other datasets for 10 epochs. Importantly, we do not use a model pre-trained on Kinetics-400 for few-shot learning on other datasets.

**Evaluation Metrics.** We evaluate our model with top-1 and top-5 accuracy on the single-label datasets. On multi-label Charades, we follow [14] to report mean average precision (mAP).

Table 3: *Comparisons on few-shot action recognition across the HMDB-51 [19], UCF-101 [34] and Kinetics-400 datasets [17] with state-of-the-art action recognition methods. We utilize ViT-B/16 as the backbone and use 8 frames for training/validation. All performances are reported as top-1 accuracy (%) in the few-shot setting using single-view inference with a spatial size of 224 × 224, where all models are directly fine-tuned from CLIP. "Avg." refers to the average performance across all datasets.*

| Method | HMDB-51 | | | | UCF-101 | | | | Kinetics-400 | | | | All |
|---|---|---|---|---|---|---|---|---|---|---|---|---|---|
| | $K$=2 | $K$=4 | $K$=8 | $K$=16 | $K$=2 | $K$=4 | $K$=8 | $K$=16 | $K$=2 | $K$=4 | $K$=8 | $K$=16 | Avg. |
| Vanilla CLIP [30] | 41.9 | 41.9 | 41.9 | 41.9 | 63.6 | 63.6 | 63.6 | 63.6 | 57.2 | 57.2 | 57.2 | 57.2 | 54.2 |
| ActionCLIP [39] | 47.5 | 57.9 | 57.3 | 59.1 | 70.6 | 71.5 | 73.0 | 91.4 | 61.0 | 63.0 | 64.8 | 68.5 | 65.5 |
| VideoPrompt [15] | 39.7 | 50.7 | 56.0 | 62.4 | 71.4 | 79.9 | 85.7 | 89.9 | - | - | - | - | 67.0 |
| X-CLIP [26] | 53.0 | 57.3 | 62.8 | 64.0 | 76.4 | 83.4 | 88.3 | 91.4 | 56.8 | 60.7 | 62.3 | 64.6 | 68.4 |
| ViFi-CLIP [31] | 57.2 | 62.7 | 64.5 | 66.8 | 80.7 | 85.1 | 90.0 | 92.7 | 37.1 | 42.8 | 49.1 | 55.5 | 65.4 |
| TC-CLIP [18] | 57.3 | 62.3 | 67.3 | 68.6 | 85.9 | 89.9 | 92.5 | 94.6 | 58.5 | 61.9 | 65.5 | 69.9 | 72.9 |
| OST [6] | 59.1 | 62.9 | 64.9 | 68.2 | 82.5 | 87.5 | 91.7 | 93.9 | 44.0 | 48.2 | 52.5 | 56.5 | 67.6 |
| Ours | **64.0** | **66.1** | **68.7** | **70.7** | **92.1** | **93.0** | **94.3** | **94.7** | **73.9** | **74.8** | **75.5** | **76.2** | **78.7** |

## 4.2 Main Results

**Action Recognition.** We compare our network against state-of-the-art methods that utilize large-scale visual pre-training and visual-text pre-training on the Kinetics-400 dataset in Tab. 1. All methods use the ViT-B/16 backbone. Using fewer input frames and views during testing, our method that employs only an 8-frame input evaluated with a single view achieves the highest top-1 and top-5 accuracy. For example, the second-best method of large-scale visual-text pre-training uses 4 temporal clips on 3 spatial crops of a video but achieves top-1 accuracy 0.5% lower than ours.

To further demonstrate the effectiveness of our framework, we conduct experiments on challenging scenarios from Kinetics-400 based on the ambiguous and non-uniform score. We select the top 10% of actions with the highest scores, representing the most difficult categories to classify. The results are shown in Tab. 9, where we achieve an average performance 2.6% higher than the second-best.

**Multi-Label Action Recognition.** We evaluate our method for multi-label action recognition on the Charades dataset in Tab. 2a. The mAPs of state-of-the-art methods are reported, and we follow [9] to use 16 frames. Our method finds 51.1 mAP, achieving 0.7 mAP more than the second-best.

**Zero-shot Action Recognition.** Our method is trained with supervision from texts and can be used for zero-shot action recognition. We compare our network with methods that adapt from CLIP, which is pre-trained on images, and with zero-shot methods developed for videos on the UCF-101 and HMDB-51 datasets in Tab. 2b. All methods use a single view during testing. Our method demonstrates superior generalization capabilities in zero-shot action recognition. Specifically, versus the latest zero-shot methods developed for video [38], our zero-shot performance on the HMDB51 dataset is 9.5% higher than 47.1%. Further highlighting the versatility of our pipeline.

**Few-shot Action Recognition.** We explore 2, 4, 8, and 16-shot action recognition on the HMDB-51 [19], UCF-101 [34], and Kinetics-400 datasets [17] in Tab. 3. With a limited amount of videos, our approach that decomposes a video action into atomic actions exhibits the highest performance. Specifically, the average performance of our approach across all shots is 78.7%, which is 5.8% higher than the second best method (TC-CLIP, with 72.9%).

## 4.3 Ablation Studies and Analysis

We perform ablation studies with the Kinetics-400 dataset to examine our approach.

**Number of Sub-texts.** Fig. 7a illustrates the number $N$ of sub-texts used in our model. As the $N$ increases from 2 to 5, the recognition accuracy of our method improves. However, we observe that the performance gain is minimal when increasing the number of sub-texts from 4 to 5. Considering both computational complexity and recognition accuracy, we opted for 4 sub-texts in our experiments.

**Correlation of TPP and Performance.** In Fig. 7b, 11 sub-text groups are generated, and we show the relationship between TPP and the action recognition performance, where TPP is

Table 4: *Comparison of Top-1 accuracies for some representative actions with N=2 and N=5 sub-actions. The Difference column shows the change in Top-1 accuracy from N=2 to N=5.*

| Action | N=2 | N=5 | Difference | Action | N=2 | N=5 | Difference |
|---|---|---|---|---|---|---|---|
| Dribbling basketball | 89.2 | 94.6 | +5.4 | Arm wrestling | 100.0 | 100.0 | 0.0 |
| Kicking soccer ball | 71.0 | 76.3 | +5.3 | Christmas tree | 100.0 | 100.0 | 0.0 |
| Lunge | 75.6 | 82.2 | +6.6 | Brushing teeth | 89.5 | 89.5 | 0.0 |
| High kick | 35.1 | 40.5 | +5.4 | Slicing onion | 98.6 | 97.4 | -1.2 |
| Clay pottery making | 92.1 | 94.7 | +2.6 | Clapping | 47.2 | 50.0 | -2.8 |

computed as the average of $TPP_c$ across all action categories. We observe a positive correlation, as shown by the fitted dashed line, with an $r^2$ value of 0.79, demonstrating the effectiveness of our TPP method in selecting sub-texts without the need for computationally intensive experiments.

**Comparison of Sub-text Generators.** Tab. 2d compares sub-texts generated by GPT-Davinci [4], GPT-3.5, GPT-4o, and LLava-v1.6-34b [22]. GPT-4o achieveshe highest TPP (60.8), along with the best Top-1 (84.52%) and Top-5 (96.7%) accuracies. Both LLaVA-1.6-34B and GPT-Davinci show similar performance, with minor differences in accuracy and TPP. These results indicate that GPT-4o provides the most effective sub-texts for action recognition.

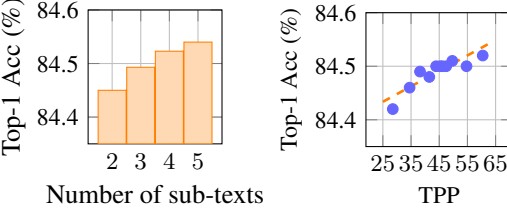

(a) The impact of numbers of sub-texts ($N$).

(b) Study the relationship between TPP and Top-1.

Figure 7: *Analysis of sub-texts.*

**Effectiveness of Components.** We study the effective usage of our coarse video embedding and fine-grained video embedding in Tab. 2e. A CLIP model is fine-tuned on the Kinetics-400 dataset to establish a performance baseline. Following the common practice of existing works [48], we build a baseline (Temporal) using a 6-layer Transformer encoder with position embedding for sequence features. Individually, both the coarse and fine-grained video embeddings enhance performance relative to the baseline; together, they achieve consistently superior results. For example, using all embeddings, we achieve a top-1 accuracy of 84.5%, which is 4.6% higher than the baseline.

**Different sub-actions.** We previously observed that the accuracy of different action categories changes with respect to the number of sub-actions (N). Taking N = 2 and N = 5 as examples, the accuracy differences across representative actions are shown in Tab. 4. The results indicate that complex, multi-stage action categories (e.g., 'Dribbling basketball' and 'Clay pottery making') benefit from large N, whereas repetitive or simple motions such as 'Clapping' and 'Slicing onion' get better accuracies with small N. We had investigated overall performance gains by using class-aware N, and only found marginal improvement over fixed N=5, 0.5% and 0.2% for Top-1 and Top-5, respectively. Given the computational cost of involving a class-aware value of N for each action, we choose to use fixed N for efficiency.

**Generalization.** To validate the generalization of our components of fine-grained video embedding structure, we apply them to state-of-the-art action recognition methods, as shown in Tab. 2c. Specifically, we study VideoPrompt [15], ATM [46] and BIKE [48]. The results indicate that we enhance the accuracy of these state-of-the-art methods, such as VideoPrompt, which shows a 2.4% increase in accuracy on the Kinetics-400 dataset.

**More.** We provide additional implementation details, comparisons, and ablation studies in the supplement, *e.g.*, analyses of few-shot, zero-shot, and supervised learning on other backbones.

## 5   Conclusion

In this paper, we propose a framework to transfer CLIP trained on image-text pairs to video action recognition. Similar to how a video forms a video action by performing a sequence of atomic actions, our key insight is to decompose a video action into atomic action descriptions using a pre-trained LLM. We then select these atomic action descriptions with a proposed metric. The global and atomic action descriptions are used to identify salient video frames from ambiguous and non-uniform videos for action recognition. Experiments on standard benchmark datasets demonstrate that our method significantly outperforms previous works in supervised, few-shot, and zero-shot settings.

## Acknowledgments and Disclosure of Funding

This work was supported by the National Natural Science Foundation of China under grant No. 62302045, and the Beijing Institute of Technology Special-Zone.

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

# Supplementary Material

We provide content omitted from the main paper due to space limitations. Specifically, the supplementary material is organized into three sections: (i) additional details for experiments and implementations, (ii) comparisons with other state-of-the-art methods in various settings, and (iii) additional ablation studies to analyze our model designs.

## A  Additional Experimental Setup

### A.1  Datasets

We describe the dataset statistics of the Kinetics-400 [17], HMDB-51 [19], UCF-101 [34], and Charades [33] datasets. The Kinetics-400 dataset is curated from YouTube, spans 400 action classes, and contains 240,000 training videos and 20,000 validation videos. The duration of each video is 10 seconds. The HMDB-51 dataset consists of 6,766 videos categorized into 51 action classes, with 3,570 videos used for training, and 1,530 videos used for testing. Each video is sourced and truncated from movies, online videos, and public databases. The UCF-101 dataset collected 9,537 training videos and 3,783 testing videos from YouTube, across 101 action categories. The Charades dataset comprises 9,848 video clips that span 157 action categories. Each video is recorded in a home environment, performed by an actor, and includes multiple human actions with temporal action annotations.

Table 5: *The hyperparameters of our model for supervised and few-shot learning.*

|  | Fully-sup | Few-shot |
|---|---|---|
| *Optimisation* | | |
| Optimizer | AdamW | |
| Optimizer betas | (0.9, 0.999) | |
| Batch size | 256 | 256 |
| Learning rate schedule | cosine | |
| Linear warmup epochs | 5 | |
| Base learning rate | 5e-6 | 5e-6 |
| Epochs | 30 | 2,10 |
| *Data augmentation* | | |
| RandomFlip | 0.5 | |
| Resize | RandomSizedCrop | |
| Crop size | 224(Default) | |
| GrayScale | 0.2 | |
| *Other regularisation* | | |
| Weight decay | 0.02 | |

### A.2  Ambiguous Score *vs.* Non-uniform Score

We describe computations of the ambiguous score and non-uniform score for Fig. 2 in the main paper. Our method does not require ground truth action labels of videos, and only uses the CLIP, for the calculation.

**Ambiguous Score.** We measure the ambiguous scores of videos by their class similarity measured with the textual encoder of CLIP. Benefiting from training on large-scale image and text pairs, the embedding space of CLIP is a compression of the world that captures the similarity between action classes. Following the global text prompt from the main paper, all action classes are extracted into $\{\mathbf{t}_c^{cls}\}_{c=1}^C$ with the textual encoder of CLIP. The similarity $\sigma_{c,c'}$ between action class c and prediction $c'$ is calculated and normalized,

$$\sigma_{c,c'} = \frac{\text{sim}(\mathbf{t}_c^{cls}, \mathbf{t}_{c'}^{cls}) + 1}{2} \ . \tag{15}$$

Table 6: *The question prompts for GPT-Davinci [4], GPT-3.5, GPT-4o, LLaMA2 [36], Phi-3 [1], and LLava-v1.6-34b [22]. Each row is a question prompt. All LLMs use the four prompts to generate sub-texts for selection.*

[This action is a sequence of four steps. For each step, describe it in detail with 'This action is' followed by a specific description of that step. Ensure each step is distinct and crucial to the action.]

[This action is defined by four unique movements. Start each movement with 'This action is' and emphasize the diversity and significance of each movement. Conclude with a brief summary that ties all movements together.]

[This action is composed of four critical phases. Each phase should start with 'This action is' and focus on different aspects of the action. Highlight the importance of each phase and avoid repetition.]

[This action is best understood by breaking it down into four concise descriptions. Begin each description with 'This action is' and make sure each one covers a different perspective of the action. End with a summarizing statement.]

Table 7: *The system prompt. We use the same system prompt for GPT-Davinci [4], GPT-3.5, GPT-4o, LLaMA2 [36], Phi-3 [1], and LLava-v1.6-34b [22].*

[You are an expert assistant in action recognition. Respond to the user's input by providing accurate, concise, and informative descriptions of the actions in a structured manner.]

For each action class c, we select the top-10 most similar other actions, and get the ambiguous scores $\text{AS}_c$ by averaging the similarity scores,

$$\text{AS}_c = \sum_{c' \in \text{top-10}_c} \sigma_{c,c'} , \tag{16}$$

where $\text{top-10}_c$ find index of the top-10 similarity action class for the action class c. In Fig. 2 of the main paper, we normalize the similarity score $\text{AS}_c$ from 1 (least ambiguous) to 10 (most ambiguous) for visualization.

Ambiguous action classes usually share different atomic actions described by the sub-texts. Thus, using the global text to identify prominent frames corresponding to the video class semantic causes ambiguity. While avoiding granularity discrepancy between the global text and video frame (Sec. Intro), we aim to decrease the granularity of the global text by augmenting it with sub-texts for ambiguous video recognition. On the ViT-B/16 backbone, we found that augmented global text achieved an accuracy of 84.5% on the 1×1 view, which is a 0.5% improvement over the 84.0% of global text, when the fine-grained branch is included.

**Non-uniform Score.** We measure the non-uniform score of a video by calculating the agreements between predictions of each frame. Leveraging a CLIP, we obtain predictions of each frame and use the global text format and textual encoder of CLIP to extract embedding $\{\mathbf{t}_{c^{\text{pred}},l}\}_{l=1}^{L}$ for all frames. The agreements between frame l and frame $l'$ are measured by similarity between the embedding, and are normalized,

$$\sigma_{l,l'} = \frac{\text{sim}(\mathbf{t}_{c^{\text{pred}},l}^{\text{cls}}, \mathbf{t}_{c^{\text{pred}},l'}^{\text{cls}}) + 1}{2} . \tag{17}$$

The non-uniform score NUS of the video is an average of the agreements between frames,

$$\text{NUS} = \sum_{l=1}^{L} \sum_{l'=1, l' \neq l}^{L} \sigma_{l,l'}. \tag{18}$$

In Fig. 2 of the main paper, we normalize the non-uniform score from 1 (uniform) to 10 (non-uniform) for visualization.

Table 8: *Performance comparison of our coarse, fine-grained, and fused video embeddings over a single-granularity method, across different levels of ambiguous and non-uniform scores, on the Kinetics-400 dataset. "Ambi" and "Non" represent ambiguous score and non-uniform score, respectively. All numbers are Top-1 accuracy (%).*

| Method | Type | Degree | | | | | | | | | |
|---|---|---|---|---|---|---|---|---|---|---|---|
| | | 1 | 2 | 3 | 4 | 5 | 6 | 7 | 8 | 9 | 10 |
| **Single** | **Ambi** | 93.0 | 90.3 | 88.3 | 84.0 | 85.7 | 82.7 | 83.0 | 76.8 | 77.2 | 69.3 |
| | **Non** | 86.5 | 84.9 | 84.6 | 85.4 | 83.9 | 82.2 | 80.8 | 81.9 | 80.3 | 80.1 |
| **Coarse** | **Ambi** | 94.1 | 91.5 | 89.8 | 86.5 | 87.4 | 84.8 | 84.2 | 78.9 | 79.4 | 70.8 |
| | **Non** | 91.1 | 88.7 | 87.8 | 90.7 | 86.7 | 85.0 | 83.6 | 84.7 | 83.1 | 82.9 |
| **Fine** | **Ambi** | 94.2 | 91.1 | 89.5 | 86.4 | 87.3 | 84.7 | 84.1 | 78.6 | 79.1 | 71.8 |
| | **Non** | 90.0 | 87.8 | 86.6 | 87.2 | 86.3 | 83.9 | 82.9 | 84.0 | 82.4 | 82.2 |
| **Fusion** | **Ambi** | **94.8** | **91.6** | **90.3** | **87.2** | **87.6** | **85.6** | **84.7** | **79.3** | **80.6** | **72.1** |
| | **Non** | **92.9** | **89.7** | **89.7** | **88.9** | **87.7** | **85.1** | **83.7** | **84.8** | **83.2** | **83.0** |

We construct two challenging subsets of the Kinetics-400 dataset based on the ambiguous score and non-uniform score. For each criterion, we select the top 10% of actions with the highest scores, representing the most difficult categories to classify. We compare the performance of our proposed method against two representative single-granularity methods ATM [46] and BIKE [48]. The results on the two challenging subsets are presented in Tab. 9.

Table 9: *Comparison of our method, BIKE [48] and ATM [46] on two challenging subsets of Kinetics-400: "Ambi Set" and "Non Set", containing the top 10% most ambiguous and non-uniform actions, respectively.*

| Method | Ambi Set | Non Set |
|---|---|---|
| ATM ViT-B/16 [46] | 70.2 | 79.8 |
| BIKE ViT-B/16 [48] | 69.3 | 80.1 |
| **Ours ViT-B/16** | 72.2 | 83.0 |

Fig. 2 (f) presents a bar chart illustrating the accuracy improvements of our method's coarse video embedding, fine-grained video embedding, and their fusion, compared to the single-granularity method. The results presented in Tab. 8 are reported across different levels of ambiguous score and non-uniform score, showcasing the effectiveness of our method on the Kinetics-400 dataset.

## A.3 Implementation Details

We use CLIP trained from [30] in our paper. During training, the textual encoder of CLIP is frozen. We summarize the optimization and data augmentation details of our method for supervised learning

Table 10: *Comparisons of zero-shot action recognition on the UCF-101 [34] and HMDB-51 [19] datasets. The "VZ" column denotes if the method is developed for zero-shot action recognition or adapted from CLIP. Our model is based on ViT-L/14.*

| Method | VZ | UCF-101 | HMDB-51 |
|---|---|---|---|
| E2E[3] | ✓ | 44.1 | 29.8 |
| ER[5] | ✓ | 51.8 | 35.3 |
| ResT[20] | ✓ | 58.7 | 41.1 |
| X-CLIP [26] | ✗ | 72.0 | 44.6 |
| DIST [29] | ✗ | 72.3 | 55.4 |
| Vita-CLIP [43] | ✗ | 75.0 | 48.6 |
| M2-CLIP [38] | ✗ | 78.7 | 47.1 |
| BIKE ViT-L [48] | ✗ | 86.6 | 61.4 |
| Text4Vis [47] | ✗ | 85.8 | 58.1 |
| **Ours ViT-L/14** | ✗ | **87.3** | **61.9** |

Table 11: *Comparisons results on the Kinetics-400 dataset. We report the FLOPs in inference phase. "Views" indicates # temporal clip $\times$ # spatial crop. The magnitude is Million ($10^6$) for parameters (Param). All methods are based on ViT-L.*

| Method | Venue | Input | Pre-training | Top-1(%) | Top-5(%) | Views | GFLOPs | Param |
|---|---|---|---|---|---|---|---|---|
| ViViT-L/16×2 [2] | ICCV'21 | 32×320$^2$ | ImageNet-21K | 81.3 | 94.7 | 4×3 | 3992×12 | 310.8 |
| ViViT-L/16×2 [2] | ICCV'21 | 32×320$^2$ | JFT-300M | 83.5 | 95.5 | 4×3 | 3992×12 | 310.8 |
| VideoSwin-L [23] | CVPR'22 | 32×384$^2$ | ImageNet-21K | 84.9 | 96.7 | 10×5 | 2107×50 | 200.0 |
| ST-Adapter ViT-L/14 [28] | NeurIPS'22 | 32×224$^2$ | WIT-400M | 87.2 | 97.6 | 3×1 | 8248 | - |
| EVL ViT-L/14 [21] | ECCV'22 | 32×224$^2$ | WIT-400M | 87.3 | - | 3×1 | 8088 | - |
| BIKE ViT-L/14 [48] | CVPR'23 | 8×224$^2$ | WIT-400M | 86.5 | - | 1×1 | 415 | 307 |
| AIM ViT-L/14 [53] | ICLR'23 | 8×224$^2$ | WIT-400M | 86.8 | 97.2 | 3×1 | 2802×1 | 341 |
| ATM ViT-L/14 [46] | ICCV'23 | 8×224$^2$ | WIT-400M | 87.3 | 97.4 | 4×3 | 421×12 | - |
| MoTED ViT-L/14 [56] | CVPR'24 | 8×224$^2$ | WIT-400M | 87.4 | 97.8 | 3×1 | 8670 | 349 |
| **Ours ViT-L/14** | NeurIPS'25 | 8×224$^2$ | WIT-400M | **87.6** | **97.8** | **1×1** | 416 | 312 |

Table 12: *Comparisons on few-shot action recognition across the HMDB-51 [19], UCF-101 [34] and Kinetics-400 datasets [17].*

| Method | Shot | HMDB-51 | UCF-101 | Kinetics-400 |
|---|---|---|---|---|
| VideoSwin [23] | 2 | 20.9 | 53.3 | - |
| VideoPrompt ViT-B/16 [15] | 5 | 56.6 | 79.5 | 58.5 |
| BIKE ViT-L/14 [48] | 2 | 73.5 | 96.1 | 75.7 |
| BIKE ViT-L/14 [48] | 5 | 77.7 | 96.5 | 78.2 |
| OST ViT-B/16 [6] | 2 | 64.8 | 90.3 | - |
| **Ours ViT-L/14** | 2 | 74.4 | 96.5 | 76.5 |
| **Ours ViT-L/14** | 5 | **78.1** | **96.9** | **79.1** |

and few-short learning in Tab. 5. For calculating TPP, the scaling functions are $\alpha(x) = -x^2 + 1$ and $\beta(x) = x$, where $x$ is the input variable. In sub-text generation, we carefully design a list of question prompts for LLMs and select sub-texts with our TPP. We consider six LLMs: GPT-Davinci [4], GPT-3.5, GPT-4o, LLava-v1.6-34b [22], LLaMA2 [36], and Phi-3 [1]. The question prompts and the system prompt for each LLM are presented in Tab. 6 and Tab. 7, respectively.

## B    Additional Experimental Results

We compare with state-of-the-art methods with a ViT-L/14 backbone. Following the setting of the main paper, the results of zero-shot action recognition on the UCF-101 [34] and HMDB-51 [19] datasets, supervised learning on the Kinetics-400 [17] dataset are in Tab. 10, Tab. 11. In all settings, our method consistently exhibits the highest performance.

As shown in Tab. 13, we compared the impact of the generative capabilities of other large language models (LLMs) on recognition accuracy, with corresponding GLops, processing time, and accuracy. The results show that even the slowest generation speed is within 10 seconds of each action. Furthermore, LLMs with strong generative capabilities have minimal impact on our method.

Table 13: *Performance comparison of Phi-3, LLaVA, LLaMA2, GPT-3.5, and GPT-4o in terms of GLOPs, processing time, and accuracy.*

| Model | GLOPs | Time(s) | Accuracy |
|---|---|---|---|
| Phi-3-14b | 636 | 7.5 | 84.50 |
| LLaVA-1.6-34b | 8.25e4 | 5.8 | 84.35 |
| LLaMA2-7b | 814 | 2.0 | 84.47 |
| GPT-3.5 | - | 1.1 | 84.47 |
| GPT-4o | - | 4.6 | 84.52 |

The common practice [40] is followed to only add FFN layers to the video encoder, which projects the video embedding to the CLIP feature space. It is unnecessary to employ an additional FFN layer for the text embedding, as the accuracy remains 84.5% with or without it.

We explored the use of visual context in generating sub-texts and identified two major limitations of this approach. i) Generating sub-texts for each video sample is computationally expensive, while

Table 14: Sub-text descriptions for the action "Situp" across multiple video samples.

| Action: Situp | Description 1 | Description 2 | Description 3 | Description 4 |
|---|---|---|---|---|
| **Video1** | The individual is lying on a mat in a gym, beginning a sit-up with arms extended overhead. | They perform a fluid motion, raising their torso towards their legs while keeping their feet flat. | The exercise is conducted in a controlled manner, focusing on core strength and stability. | The environment suggests a well-equipped gym, appropriate for various fitness activities. |
| **Video2** | An individual is seen lying on the grass performing a sit-up exercise in an outdoor setting. | The exercise involves raising the upper body towards the knees, engaging the core muscles. | The person is wearing casual workout attire, suitable for outdoor physical activities. | The green, open space offers a natural and refreshing environment for a fitness routine. |
| **Video3** | Two individuals are in a gym; one is performing an exercise on the floor while the other is walking around, possibly coaching. | The environment indicates a focus on personal training, as evidenced by the sign in the background. | The person on the floor appears to be doing sit-ups or a similar core exercise, suggesting a targeted workout session. | The overall scene suggests a casual yet focused atmosphere, typical of a fitness or personal training session. |

class-level generation requires choosing a representative video, risking inconsistency and limited coverage. ii) For complex or ambiguous actions, visual context often introduces noise and spurious details, weakening frame alignment and degrading performance. To illustrate, take 'Situp' as an example, we randomly selected three sample videos and uniformly sampled eight frames from each. These were provided as input to GPT-4o to generate visually informed sub-texts as shown in Tab. 14. These sub-texts introduce scene-specific elements (e.g., attire, environment, presence of others) that do not directly contribute to the representation of atomic actions and can confound alignment.

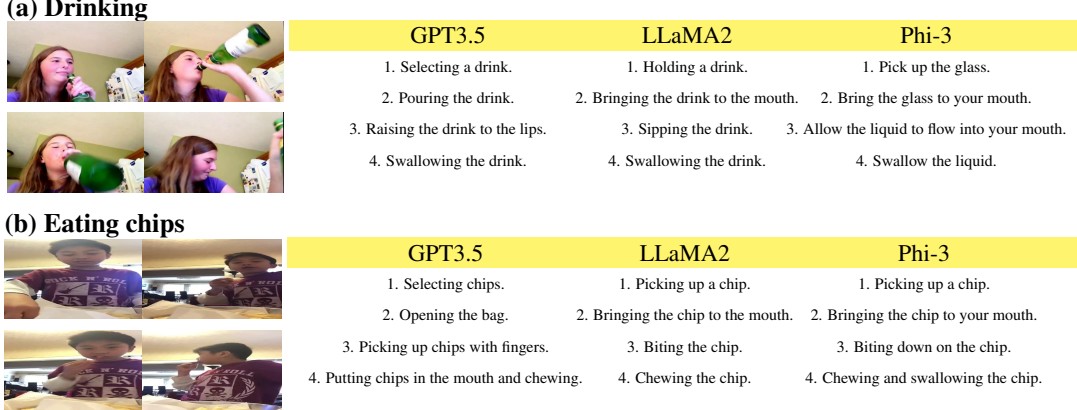

**(a) Drinking**

| | GPT3.5 | LLaMA2 | Phi-3 |
|---|---|---|---|
| | 1. Selecting a drink. | 1. Holding a drink. | 1. Pick up the glass. |
| | 2. Pouring the drink. | 2. Bringing the drink to the mouth. | 2. Bring the glass to your mouth. |
| | 3. Raising the drink to the lips. | 3. Sipping the drink. | 3. Allow the liquid to flow into your mouth. |
| | 4. Swallowing the drink. | 4. Swallowing the drink. | 4. Swallow the liquid. |

**(b) Eating chips**

| | GPT3.5 | LLaMA2 | Phi-3 |
|---|---|---|---|
| | 1. Selecting chips. | 1. Picking up a chip. | 1. Picking up a chip. |
| | 2. Opening the bag. | 2. Bringing the chip to the mouth. | 2. Bringing the chip to your mouth. |
| | 3. Picking up chips with fingers. | 3. Biting the chip. | 3. Biting down on the chip. |
| | 4. Putting chips in the mouth and chewing. | 4. Chewing the chip. | 4. Chewing and swallowing the chip. |

Figure 8: *Example sub-texts generated from GPT-3.5, LLaMA2 [36], and Phi-3 [1].*

## C    Additional Ablation Studies

**LLMs Selection.** We ablate the LLM used for sub-text generation in Tab. 7. We experiment with GPT-Davinci [4], GPT-3.5, GPT-4o, and LLava-v1.6-34b [22]. Sub-text groups generated from GPT-4o have the highest TPP and action recognition performance.

**Visualization.** Additional example sub-texts generated by GPT-3.5, LLaMA2 [36], and Phi-3 [1] are presented in Fig. 8. Furthermore, We provide more visualizations than Fig. 2 of the main paper in Fig. 9 and Fig. 10 on Kinetics-400 [17], UCF-101 [34], and HMDB-51 [19] datasets.

**Limitation and Future Work** Fine-grained action recognition is challenging, especially in complex scenarios, with no satisfactory solution even from large vision-language models like GPT-4v. Our method mitigates this by generating sub-texts once for each global action, but future work could explore more robust approaches.

**Broader Impacts** Our work advances the study of fine-grained action recognition methods and holds promise for a wide range of applications, including sports analytics, human-computer interaction, surveillance, video understanding, and related areas.

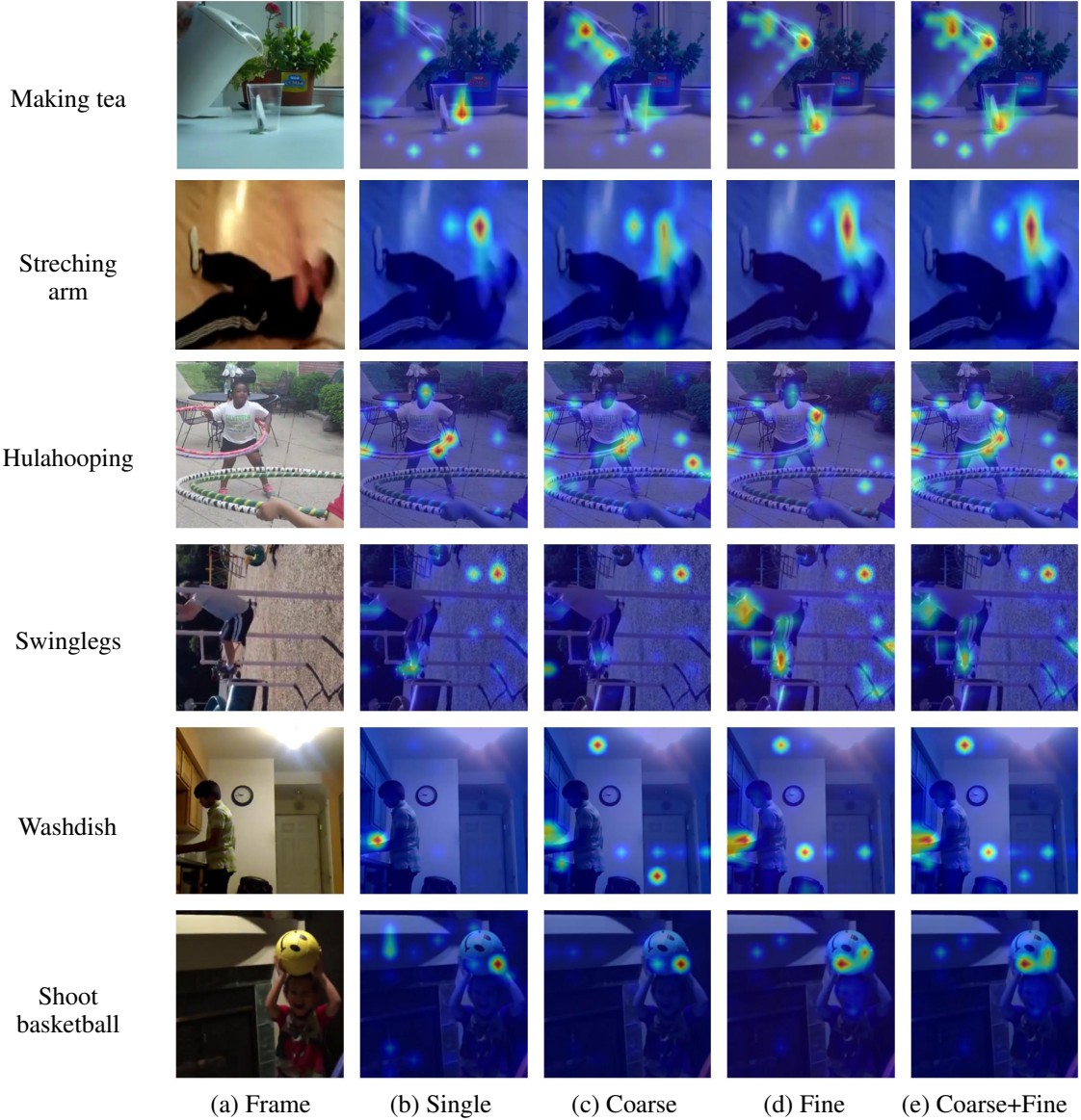

Making tea

Streching arm

Hulahooping

Swinglegs

Washdish

Shoot basketball

(a) Frame     (b) Single     (c) Coarse     (d) Fine     (e) Coarse+Fine

Figure 9: *Examples that illustrate the ineffectiveness of directly aligning video frames and global text, from (a-e) heatmaps. (a) Video frames from the Kinetics-400 dataset [17] with their class name annotated at the left. (b) Heatmaps of the single granularity-based (single) method [48], which are limited by the granularity difference between the video frame and global text. (c-e) Heatmaps using coarse video embedding, fine-grained video embedding, and the fusion of the two.*

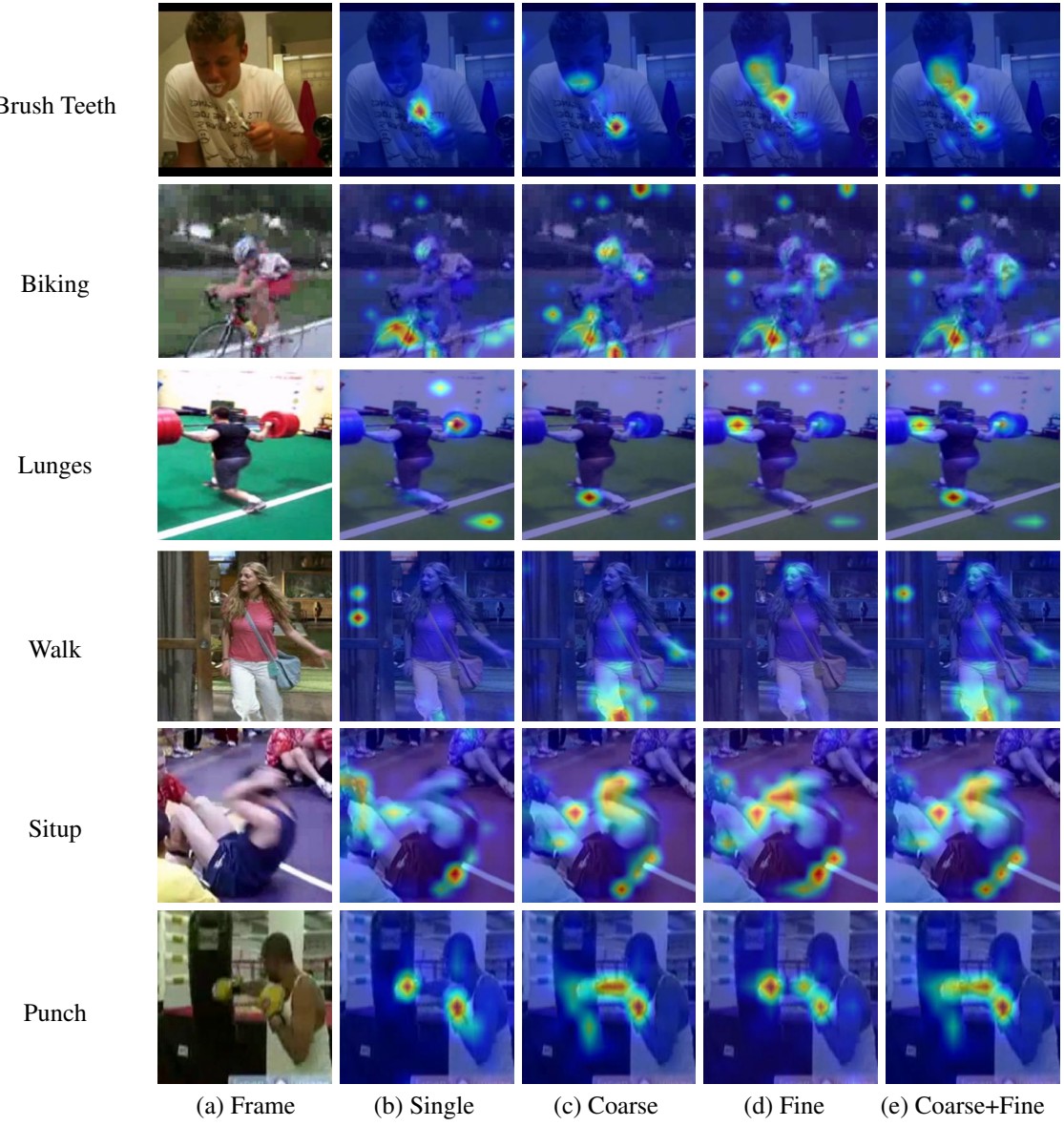

| (a) Frame | (b) Single | (c) Coarse | (d) Fine | (e) Coarse+Fine |

Figure 10: *Examples that illustrate the ineffectiveness of directly aligning video frames and global text, from (a-e) heatmaps. (a) Video frames from the UCF-101 dataset [34] and HMDB-51 dataset [19] with their class name annotated at the left. (b) Heatmaps of the single granularity-based (single) method [48], which are limited by the granularity difference between the video frame and global text. (c-e) Heatmaps using coarse video embedding, fine-grained video embedding, and the fusion of the two.*

