# OpenReview forum: "Storyboard-guided Alignment for Fine-grained Video Action Recognition"
_NeurIPS.cc/2025/Conference — NeurIPS 2025 poster_

### Official Review · Reviewer_ys6V · 2025-06-14

**Clarity:** 2
**Significance:** 2
**Originality:** 2
**Rating:** 4
**Confidence:** 3

**Summary:**

Fine-grained video understanding is a challenging problem. The authors aim to address this issue through a multi-granularity framework. Specifically, they propose storyboard-guided fine-grained video understanding  (SFAR), which generates  fine-grained (sub-text)  detailed descriptions of common atomic actions corresponding to each global semantic, leveraging large language models. Experiments were conducted on various video action recognition datasets to demonstrate the performance of the approach under different settings, including supervised, few-shot, and zero-shot scenarios.

**Questions:**

1). The main issue with the paper is that the overall contribution is incremental due to limited technical novelty. Although the authors clearly specify their primary contributions compared to existing approaches, the emphasis on the novelty of their work remains insufficient.

2). In the Global Text Augmentation module, both fine-grained (sub-text) and global context are utilized. However, the authors refer to this as global text augmentation. Could you clarify why it is named “global” when it also incorporates fine-grained elements?

3). The formulation of Fine-grained Importance and Coarse-grained Importance is not clear. This part requires a more detailed analysis and thorough discussion, especially since it represents the main contribution of the paper.

4). Some sections of the writing need significant improvement to provide clearer explanations. For example, abbreviations should be introduced before they are used. The captions of the figures are too long and need to be clearer and more focused. Detailed information from the captions can be included in the body or explanation section instead.

**Ethical Concerns:**

["NO or VERY MINOR ethics concerns only"]

**Final Justification:**

Considering the clarifications provided by the authors during the rebuttal period, some of my concerns and questions have been addressed. Thus, I am inclined to increase my score to borderline accept.

**Limitations:**

Yes

**Paper Formatting Concerns:**

No, Problem

**Quality:**

2

**Strengths And Weaknesses:**

The proposed approach addresses fine-grained video understanding through a multi-granularity framework. Specifically, the authors employ a multimodal approach that integrates coarse-grained and fine-grained representative feature learning. The problem formulation of this work is interesting, it aims to learn and understand fine-grained video or action recognition by utilizing both coarse-grained and fine-grained multimodal features. In the textual semantic branch, they use global semantic learning to capture coarse-grained features. Similarly, to learn fine-grained action features, the authors generate sub-action texts. These learned features are optimized jointly with the visual branch, encompassing both coarse-grained and fine-grained action features.

However, the novelty of the proposed approach is limited. Similar methods have been explored previously, which reduces the overall contribution of this work. For a top-tier conference like NeurIP, a more significant novel contribution is required.

---

> ### Author Rebuttal · Authors · 2025-07-31
>
> Thank you for your comments and for recognizing that the problem formulation of this work is interesting. Below, we address your concerns point by point.
>
> > **Q1: Novelty.**
>
> **A1.** Please refer to L8–14 of the abstract and L31-64 of the introduction section. 'We propose a multi-granularity framework, SFAR. SFAR generates fine-grained descriptions of common atomic actions for each global semantic using a large language model. Unlike existing works that refine global semantics with auxiliary video frames, SFAR introduces a filtering metric to ensure correspondence between the descriptions and the global semantics, eliminating the need for direct video involvement and thereby enabling more nuanced recognition of subtle actions.'
>
> > **Q2: Why Is It Named “global”?**
>
> **A2.** As detailed in Eq. (8) of the main paper, the augmented global text embedding $\hat{\mathbf{T}}\_{\text{c}}$ is constructed via a residual connection based on the original global text $\mathbf{T}\_{\text{c}}$. Compared to the fine-grained video embedding module, the coarse video embedding emphasizes holistic video semantics. Therefore, we use the term “global” to reflect the module’s target.
>
> > **Q3: Formulation of Coarse-Grained Importance and Fine-Grained Importance.**
>
> **A3.** For coarse-grained importance, we compute scores $a\_{\text{l}}^\text{coarse}$ (Eq. (9)) using the augmented global text embedding $\hat{\mathbf{T}}\_{\text{c}}$ (Eqs. (6)–(8)) and video frame embedding $\mathbf{v}\_{\text{l}}^\text{cls}$, as described in L184–189.
>
> For fine-grained importance score $a\_{\text{l}}^\text{fine}$, as defined in L211–212 and Eq. (11), we measure the maximum average similarity between $\mathbf{v}\_{\text{l}}^\text{cls}$ and sub-texts $\mathbf{S}\_{\text{c}, \text{n}}$, followed by SoftMax normalization.
>
> > **Q4: Organisation Issue.**
>
> **A4.** Our intention with the detailed figure captions was to ensure that readers could understand the core methodology and concepts even when viewing the figures independently. We will improve conciseness as suggested.

---

> > ### Comment · Reviewer_ys6V · 2025-08-05
> >
> > Thank you to the authors for their thorough response. I have carefully reviewed the authors' responses and the discussion among reviewers. Considering the clarifications provided, I am inclined to increase my score to borderline accept.
> >
> > Thank you to the authors for their detailed and constructive responses.

---

> > > ### Author Response · Authors · 2025-08-06
> > >
> > > Thank you for all the helpful comments and positive assessments.
> > >
> > > We are grateful to reviewer 'ys6V' for upgrading the score. Thank you again for your time and support.

---

### Official Review · Reviewer_DdM6 · 2025-06-25

**Clarity:** 2
**Significance:** 3
**Originality:** 3
**Rating:** 4
**Confidence:** 4

**Summary:**

This paper proposes a multi-granularity framework SFAR for fine-grained video action recognition to further explore the potential of text encoders in adapting to the confusion and unevenness of action atomic. The core theory lies in decomposing action text into a sub-text set with LLM at a fine-grained level, and using this text set to enhance visual features at multiple granularities (i.e., coarse level and fine-grain level), and finally fusing the features to predict action categories. Rich experiments on multiple benchmarks and multiple tasks have demonstrated the effectiveness of the proposed method.

**Questions:**

Although this article has designed an effective framework for fine-grained video action recognition, it suffers from deficiencies (See Weaknesses) in written expression and theoretical analysis, etc., my current rating is 3.

**Ethical Concerns:**

["NO or VERY MINOR ethics concerns only"]

**Final Justification:**

In general, this paper is valuable and promotes the research in this field. Considering the positive opinions of other reviewers, I choose to raise my rating to Borderline accept.

**Limitations:**

Yes.

**Paper Formatting Concerns:**

None.

**Quality:**

3

**Strengths And Weaknesses:**

Strengths：
1. The author conducted in-depth research on the characteristics of fine-grained actions and analyzed the corresponding issues. The proposal of sub-text generation for fine-grained video action recognition is insightful.
2. The design of the text prompt perplexity score seems effective and offers a new approach for the correlation analysis between the text generated by LLMs and video information.
3. The ablation experiment design is diverse, and extensive and effective validation has been conducted for the core components in the text.

Weaknesses:
1. I find the writing of this paper flawed, as the authors did not carefully check and optimize the content before submission. (1) The figure captions are overly verbose (even containing extensive symbolic formulas), making it difficult for readers to grasp the key theoretical points. (2) There is redundancy between the textual descriptions and equations in the theoretical presentation, such as in Lines 140-144 and Lines 193-197. (3) The use of embedded formatting for multiple figures creates a sense of disorganization for readers.
2. The limitations of the text prompt perplexity score. The author did not provide specific analysis and explanations for the design of the text prompt perplexity score. For example, what were the design motivations for each factor in Equation 3? What is the theoretical impact of the values on the quality of text features? How was TTP specifically utilized during the enhancement process?
3. The generated sub-text remains in the form of verb-noun. However, the encoder of CLIP focuses on static images. How can SFAR ensure the adaptability of the generated text to the dynamic video information?

---

> ### Author Rebuttal · Authors · 2025-07-31
>
> Thank you for your valuable comments and for recognizing that our ablation experiment design is diverse. Below, we address your concerns point by point.
>
> > **W1: Content Optimization.**
>
> **A1.** (1) We intentionally designed captions to be comprehensive and self-contained, ensuring symbolic formulas and context are immediately accessible without requiring readers to cross-reference text sections; (2) Thanks for the catch and fixed; (3) We will try to leave more room for figures.
>
> > **W2: Motivations of Text Prompt Perplexity Score.**
>
> **A2.** The Text Prompt Perplexity (TPP) score (Eq. (3) in the main paper) is designed with two key factors: similarity with global text and diversity among sub-texts.
> \begin{align}
>     &\text{TPP}\_\text{c} = \exp \bigg( - \frac{1}{N} \sum\_{\text{n}=1}^{N} \log \big( \alpha (\sigma\_{\text{c},\text{n}} ) \beta(\delta\_{\text{c},\text{n}}) \big) \bigg) \ .
>     \tag{3}
> \end{align}
> First, we employ $\sigma\_{\text{c},\text{n}}$ to quantify the similarity between a global text and its corresponding sub-texts, a higher value of $\sigma\_{\text{c},\text{n}}$ indicates greater similarity. Next, $\delta\_{\text{c},\text{n}}$ is used to evaluate the similarity among sub-texts themselves, where a higher value of $\delta\_{\text{c},\text{n}}$ indicates greater diversity across the sub-texts. The linear scaling functions $\alpha(\cdot)$ and $\beta(\cdot)$ ensure that generated sub-texts with higher similarity and diversity scores receive higher TPP scores.
>
> > **W3: Video Alignment.**
>
> **A3.** Although the sub-texts generated by SFAR adopt a verb-noun structure, they are not bound to static interpretations of a single frame. As described in Eq. (11), SFAR computes similarities between each sub-text and individual frames within the CLIP embedding space, enabling the identification of frame-level visual evidence corresponding to atomic actions. These interactions are then aggregated through temporal pooling (Eq. (12)), which captures inter-frame dependencies and facilitates the flow of contextual information over time. Additionally, SFAR incorporates a 6-layer Transformer module within its visual encoder to explicitly model the temporal progression of dynamic content. This further enriches the representation of motion and continuity across the video sequence.
>
> Through these mechanisms, SFAR effectively ensures the adaptability of the generated text to the dynamic video information.

---

> ### Comment · Reviewer_DdM6 · 2025-08-06
> **Official Comment by Reviewer DdM6**
>
> Thank for the authors' efforts in rebuttal. This has resolved my concerns.
> 1. I personally do not agree with the author's view on the writing mode, as it has had a negative impact on my reading. This is in line with the opinion of Reviewer ys6V. I hope that this will be reflected in the revised version.
> 2. The author resolved my confusion about the generation principle of Sub-text, which is highly commendable.
> 3. The authors' theoretical illustration of video-text cross-modal alignment does not convince me that, even for a single video frame, CLIP still has difficulty focusing on verb-image correlations in action primitives because its feature space is noun-dominated.
>
> In general, this paper is valuable and promotes the research in this field. Considering the positive opinions of other reviewers, I choose to raise my rating to Borderline accept.

---

> > ### Author Response · Authors · 2025-08-06
> >
> > Thank you for the comments. We will try our best to improve the readability of the paper, the clarity of TPP score and video alignment in the revised version.

---

### Official Review · Reviewer_VmYv · 2025-07-01

**Clarity:** 3
**Significance:** 2
**Originality:** 2
**Rating:** 4
**Confidence:** 4

**Summary:**

The paper proposes SFAR, a multi-granularity framework for fine-grained video action recognition. By generating atomic-level sub-texts using large language models, it improves video-text alignment and achieves strong performance across supervised, zero-shot, and few-shot settings.

**Questions:**

See weakness.

**Ethical Concerns:**

["NO or VERY MINOR ethics concerns only"]

**Final Justification:**

My concerns have been addressed. I keep my rating at "Borderline accept"

**Limitations:**

The authors mentioned the limitations in the supplementary material.

**Paper Formatting Concerns:**

There are no formatting issues.

**Quality:**

3

**Strengths And Weaknesses:**

**Strengths**
1. The paper is generally well-written and easy to follow.
2. The issue of granularity mismatch between videos and texts is well recognized and analyzed. The proposed SFAR seems to be effective on this issue.
3. Extensive experiments across multiple benchmarks (supervised, few-shot, and zero-shot) show strong quantitative results.

**Weaknesses**
1. The fine-grained video embedding relies on sub-texts generated by an LLM based solely on the class name, without incorporating any visual context. However, these sub-texts may miss essential atomic actions or include redundant or vague descriptions, which can limit the effectiveness of frame-level alignment.

2. Both the coarse and fine-grained embeddings are derived from the same frame-level features. While they are guided by different text features, they may capture overlapping information, resulting in redundant representations. This raises concerns about whether the added architectural complexity of dual embeddings is always justified by meaningful performance gains.

3. The proposed sub-text generation strategy is well-motivated; however, including more detailed ablations—such as comparisons with random sub-texts or alternative prompt templates—would better highlight the effectiveness of the TPP-based selection process.

---

> ### Author Rebuttal · Authors · 2025-07-31
>
> Thank you for your insightful comments and for acknowledging that our paper is generally well-written and easy to follow. Below, we address your concerns point by point.
>
> > **W1: LLM-Generated Sub-texts Without Visual Context.**
>
> **A1.** We did explore the use of visual context previously and found two major limitations for this solution.
>
> 1. Generating sub-texts for each individual video sample entails significant computational and time costs. Alternatively, generating sub-texts at the class level introduces the challenge of selecting a representative video, which can affect consistency and coverage across diverse instances.
>
> 2. For ambiguous, non-uniform, or contextually intricate actions, incorporating visual context yields sub-texts dominated by noisy or spurious information. This undermines frame-level alignment and degrades model performance.
>
> To illustrate, take 'Situp' as an example, we randomly selected three sample videos and uniformly sampled eight frames from each. These were provided as input to GPT-4o to generate visually informed sub-texts:
>
> |      'Situp'   |    |      |     |        |
> |--------------------------|------|------|------|------|
> | Video1 |He individual is lying on a mat in a gym, beginning a sit-up with arms extended overhead.|They perform a fluid motion, raising their torso towards their legs while keeping their feet flat.|The exercise is conducted in a controlled manner, focusing on core strength and stability.|The environment suggests a well-equipped gym, appropriate for various fitness activities.|
> | Video2 |An individual is seen lying on the grass performing a sit-up exercise in an outdoor setting.|The exercise involves raising the upper body towards the knees, engaging the core muscles.|The person is wearing casual workout attire, suitable for outdoor physical activities.|The green, open space offers a natural and refreshing environment for a fitness routine.|
> | Video3 |Two individuals are in a gym; one is performing an exercise on the floor while the other is walking around, possibly coaching.|The environment indicates a focus on personal training, as evidenced by the sign in the background.|The person on the floor appears to be doing sit-ups or a similar core exercise, suggesting a targeted workout session.|The overall scene suggests a casual yet focused atmosphere, typical of a fitness or personal training session.|
>
> These sub-texts introduce scene-specific elements (e.g., attire, environment, presence of others) that do not directly contribute to the representation of atomic actions and can confound alignment.
>
> In contrast, our proposed sub-text generation method is guided by the Text Prompt Perplexity (TPP) score, which promotes relevance by emphasizing informative atomic actions. To further minimize ambiguity and redundancy, we employ an attention-based alignment mechanism (Eqs. (6–8) and (11)) that assigns lower weights to less informative content, yielding sub-texts that are both precise and semantically aligned with frame-level dynamics.
>
> > **W2: Dual Embeddings.**
>
> **A2.** We demonstrate the complementary nature of the coarse and fine-grained embeddings through corresponding heatmaps. As visualized in Fig. 2, the attention maps for coarse, fine, and their combination show distinct focus regions across three examples: 'Making a sandwich', 'Pull up', and 'Shearing sheep'.
> For instance, in 'Shearing sheep', the coarse branch (Fig. 2c) primarily attends to the head of the sheep and the area where shearing occurs on the right, while the fine branch (Fig. 2d) highlights the body of the sheep, the shearing tool, and the wool falling to the ground. The combined attention (Fig. 2e) effectively integrates both perspectives.
>
> These visualizations confirm that coarse and fine embeddings capture different aspects of the video, thus mitigating redundancy and supporting the effectiveness of the dual-embedding design. Quantitative results are shown in Tab. 2e.
>
> > **W3: Ablations of Alternative Prompt Templates.**
>
> **A3.** Please kindly refer to Fig. 7b of the main paper, where we present the relationship between TPP scores and the corresponding action recognition accuracy of sub-texts generated using alternative prompt templates. Representative examples of these templates are provided in Tab. 5 of the supplementary material. These results clearly demonstrate that prompts yielding higher TPP scores also lead to superior recognition performance, validating the effectiveness of our TPP-based selection strategy. We will highlight these comparisons more explicitly in the revised version for added clarity.

---

### Official Review · Reviewer_LH8r · 2025-07-03

**Clarity:** 2
**Significance:** 3
**Originality:** 3
**Rating:** 4
**Confidence:** 3

**Summary:**

This paper introduces Storyboard-guided Fine-grained Action Recognition (SFAR), a multi-granularity framework for fine-grained video action recognition.
The key innovation is using LLMs to decompose class-level “global” action labels into multiple atomic sub-actions. Then, these sub-actions are used as fine-grained prompts to identify and aggregate class-discriminative frames using CLIP’s multimodal embedding space. Finally, the method fuses both coarse and fine-grained video embeddings for classification.
Empirical results on various datasets demonstrate improvements over state-of-the-art in supervised, few-shot, and zero-shot settings.

**Questions:**

1. Regarding the fixed number of sub-actions (N): In your experiments where you varied the fixed number of sub-actions (N), did you observe that the accuracy for different action categories changed differently? In other words, are there specific action classes that benefit from a larger or smaller N, and how does this variation affect overall recognition performance?

2. Lack of Sequential Diversity in Sub-action Decomposition: Since the method produces only a single sub-action sequence per action, how does it handle cases where the same coarse action can be performed with different valid temporal orderings?

3. Absence of Temporal Modeling: Given that your framework does not explicitly model temporal dependencies, how does it capture critical temporal dynamics, especially for actions where the order of sub-actions is essential?

**Ethical Concerns:**

["NO or VERY MINOR ethics concerns only"]

**Final Justification:**

Thanks for the author's response, it resolved my almost concerns, I will keep my positive rating for "Borderline accept".
For the fixed number N of sub-actions ablations, it is better to be considered in the final version, as it is an important discussion for the reproducibility and motivation for the design.

**Limitations:**

yes

**Quality:**

3

**Strengths And Weaknesses:**

Strengths:

1.	Clear Motivation. Clearly articulates the limitations of prior CLIP-based and vision-language models in capturing atomic-level video actions, with strong illustrative examples for both ambiguous and non-uniform action distributions.

2.	Multi-Granularity Embedding: Presents a novel framework for fusing coarse (global augmented with sub-texts) and fine-grained (atomic sub-text-aligned) video embeddings. The network design matches with the motivation, targeting the granularity mismatch problem.

Weakness:

1. Fixed Number of Sub-action for various coarse action: The method sets a fixed number of sub-actions for all action classes, regardless of action complexity or video duration. This one-size-fits-all approach is suboptimal, as some simple actions require only a single atomic step, while complex actions in longer videos may need a finer and variable decomposition.

2. Lack of Sequential Diversity in Sub-action Decomposition: The approach generates a single, fixed sub-action sequence per coarse action, overlooking that the same action can correspond to multiple valid temporal orderings of sub-actions. This limits the model's ability to accommodate intra-class variability in action execution.

3. Absence of Temporal Modeling: The framework does not explicitly model temporal relationships or sequence dependencies between video frames. Treating videos as unordered frame sets may miss important temporal dynamics necessary for accurate action recognition.

---

> ### Author Rebuttal · Authors · 2025-07-31
>
> We appreciate your insightful comments and your recognition of our novel multi-granularity framework. We address your concerns point by point below.
> > **Q1: Fixed Number of Sub-Actions (N).**
>
> **A1.** We previously observed that the accuracy of different action categories changes with respect to the number of sub-actions (N). However, we set a fixed N, rather than using a class-aware N. The reason is given below.
>
> Taking N = 2 and N = 5 as examples, the accuracy differences across representative actions are given below.
>
> | Action                   | N=2  | N=5  | Difference | Action                          | N=2  | N=5  | Difference |
> |--------------------------|------|------|------------|----------------------------------|------|------|------------|
> | Dribbling basketball     | 89.2 | 94.6 | +5.4       | Arm wrestling                   | 100.0| 100.0| 0.0        |
> | Kicking soccer ball      | 71.0 | 76.3 | +5.3       | Decorating the Christmas tree   | 100.0| 100.0| 0.0        |
> | Lunge                    | 75.6 | 82.2 | +6.6       | Brushing teeth                  | 89.5 | 89.5 | 0.0        |
> | High kick                | 35.1 | 40.5 | +5.4       | Slicing onion                   | 98.6 | 97.4 | -1.2       |
> | Clay pottery making      | 92.1 | 94.7 | +2.6       | Clapping                        | 47.2 | 50.0 | -2.8       |
>
> The results indicate that complex, multi-stage action categories (e.g., 'Dribbling basketball' and 'Clay pottery making') benefit from large N, whereas repetitive or simple motions such as 'Clapping' and 'Slicing onion' get better accuracies with small N.
>
> We had investigated overall performance gains by using class-aware N, and only found marginal improvement over fixed N=5, 0.5\% and 0.2\% for Top-1 and Top-5, respectively.
>
> Given the computational cost of involving a class-aware value of N for each action, we choose to use fixed N for efficiency.
>
> > **Q2: Sequential Diversity in Sub-Action Decomposition.**
>
> **A2.** While our method adopts a fixed sub-action sequence for each coarse action, it does not rely on this ordering for accurate recognition. In fact, our model is designed to remain robust to different valid temporal arrangements of sub-actions through the use of temporally agnostic attention mechanisms.
>
> Specifically, the fine-grained importance score $a\_{\text{l}}^\text{fine}$ is computed independently for each frame $\mathbf{v}\_{\text{l}}^\text{cls}$ by measuring its maximum average similarity with all sub-texts $\mathbf{S}\_{\text{c}, \text{n}}$. This design allows the model to capture fine-grained relevance without relying on a fixed temporal sequence of atomic actions. Similarly, the coarse-grained importance score $a\_{\text{l}}^\text{coarse}$ leverages the augmented global text embedding $\hat{\mathbf{T}}\_{\text{c}}$, which is constructed to be invariant to sub-action order. Together, these components ensure that both local and global relevance signals are captured without enforcing a fixed temporal sequence.
>
> To validate this, we randomly shuffled the order of the sub-texts and observed that the top-5 accuracy remained unchanged, while the top-1 accuracy decreased marginally by only 0.01.
>
> > **Q3: Absence of Temporal Modeling.**
>
> **A3.** To address the absence of explicit temporal modeling, we construct a baseline (Temporal) using a 6-layer Transformer encoder to capture sequence dependencies among video frames (Line 300, main paper). To effectively model critical temporal dynamics, especially for actions reliant on sub-action ordering, we compute fine-grained and coarse-grained importance scores $a\_{\text{l}}^\text{fine}$ and $a\_{\text{l}}^\text{coarse}$, respectively, which are adaptively aligned with sub-action semantics. These scores modulate the corresponding frame embeddings $\mathbf{v}\_{\text{l}}^\text{cls}$ without disrupting the video’s inherent temporal structure. The resulting weighted embeddings are then aggregated and fused via two feedforward layers (Eq. 13), enabling the model to integrate multi-granularity information and produce a temporally adaptive video-level representation.

---

> ### Comment · Reviewer_LH8r · 2025-08-08
>
> Thanks for the author's respones, it resovled my almost concerns, I will keep my positive rating.
> For the fixed number N of sub-actions ablations, it is better to be considered in the final version,
> as it is an important discussion for the reproducbility and motivation for the design.

---

> > ### Author Response · Authors · 2025-08-08
> >
> > We appreciate your valuable comments. We will certainly include it in the revised version.

---

### Decision · Program_Chairs · 2025-09-17

**Decision:**

Accept (poster)

**Comment:**

This paper introduces a multi-granularity framework for fine-grained video understanding. Fine-grained descriptions of common atomic actions are generated using LLMs and employed as prompts to identify and aggregate class-discriminative frames within CLIP’s multimodal embedding space. The framework demonstrates strong performance across supervised, zero-shot, and few-shot settings.

All reviewers found the problem well-motivated and the proposed design intuitive. Reviewer concerns were adequately addressed in the rebuttal, and thus the AC concurs with the reviewers and recommends acceptance of the paper. As noted by Reviewer LH8r, the ablation on the fixed number N of sub-actions should be included in the final version, as it is an important point for both reproducibility and the motivation of the design.